

# Contrasting Local and Long-Range Transported Warm Ice-Nucleating Particles During an Atmospheric River in Coastal California, USA

Andrew C Martin[1], Gavin Cornwell[2], Charlotte Marie Beall[3], Forest Cannon[1], Sean Reilly[4], Bas Schaap[3], Dolan Lucero[2], Jessie Creamean[5,6], F. Martin Ralph[1], Hari T Mix[4], and Kimberly Prather[2,3]

[1]Center for Western Weather and Water Extremes, Scripps Institution of Oceanography, UC San Diego, La Jolla, CA
[2]Department of Chemistry and Biochemistry, UC San Diego, La Jolla, CA
[3]Scripps Institution of Oceanography, UC San Diego, La Jolla, CA
[4]Santa Clara University, Dept. of Environmental Studies and Sciences, Santa Clara, CA
[5]Cooperative Institute for Research in Environmental Sciences, University of Colorado, Boulder, CO
[6]Physical Sciences Division, National Oceanic and Atmospheric Administration, Boulder, CO

**Correspondence:** Dr. Kimberly Prather (kprather@ucsd.edu)

**Abstract.** Ice nucleating particles (INP) have been found to influence the amount, phase, and efficiency of precipitation from winter storms, including atmospheric rivers. Warm INP, those that initiate freezing at temperatures warmer than -10 °C, are thought to be particularly impactful because they can create primary ice in mixed-phase clouds, enhancing precipitation efficiency. The dominant sources of warm INP during atmospheric rivers, the role of meteorology in modulating transport and injection of warm INP into atmospheric river clouds and the impact of warm INP on mixed-phase cloud properties are not

5  well-understood. Time-resolved precipitation samples were collected during an atmospheric river in Northern California, USA during winter 2016. Precipitation was collected at two sites, one coastal and one inland, that are separated by less than 35 km. The sites are sufficiently close that airmass sources during this storm were almost identical, but the inland site was exposed to terrestrial sources of warm INP while the coastal site was not. Warm INP were more numerous in precipitation at the inland

10  site by an order of magnitude. Using FLEXPART dispersion modelling and radar-derived cloud vertical structure, we detected influence from terrestrial INP sources at the inland site, but did not find clear evidence of marine warm INP at either site. We episodically detected warm INP from long-range transported sources at both sites. By extending the FLEXPART modelling using a meteorological reanalysis, we demonstrate that long-range transported warm INP are observed only when the upper tropospheric jet provided transport to cloud tops. Using radar-derived hydrometeor classifications, we demonstrate that hy-

15  drometeors over the terrestrially-influenced inland site were more likely to be in the ice phase for cloud temperatures between 0 °C and -10 °C. We thus conclude that terrestrial and long-range transported aerosol were important sources of warm INP during this atmospheric river. Meteorological details such as transport mechanism and cloud structure were important in determining warm INP source strength and injection temperature, and ultimately the impact of warm INP on mixed phase cloud properties.



## 1 Introduction

Atmospheric Rivers (ARs) are responsible for significant precipitation in many extratropical regions (Ralph et al., 2006; Neiman et al., 2011; Ralph and Dettinger, 2012; Dettinger, 2013; Lavers and Villarini, 2013). On the windward side of some continents, including in the US state of California, ARs are responsible for up to 50% of the annual rainfall (Dettinger et al., 2011; Lavers and Villarini, 2015) It has long been known that naturally occurring tropospheric aerosols can influence precipitation by serving as heterogeneous ice nucleating particles (INPs) (Vali, 1971; Pitter and Pruppacher, 1973; Maki et al., 1974; DeMott et al., 2011). INPs may also influence precipitation from ARs. Ault et al. (2011) compared two dynamically similar ARs that impacted California and found that precipitation residues classified as dust or biological were more plentiful in the AR that produced more precipitation and more mountain snow. By extending similar analyses, Creamean et al. (2013) showed a relationship between the amount of dust and biological precipitation residues and the precipitation amount and phase. Creamean et al. (2013, 2015) also found that precipitation occurring after the storm's cold front passed was more enriched in these residue types. Numerical weather prediction experiments (Fan et al., 2014) have demonstrated that dust aerosols can invigorate precipitation in California AR by enhancing snow formation in mixed-phase orographic clouds.

Several studies have suggested that long-range transported dust aerosols are often mixed with biological remnant material (Conen et al., 2011; Murray et al., 2015; O'Sullivan et al., 2016). The source of the remnant material may allow dust/biological mixtures to serve as "warm" INPs. Herein, we define warm INPs as particles that cause freezing of supercooled liquid cloud droplets through immersion nucleation at temperatures warmer than -10 °C (Stopelli et al., 2015). Several other types of biological aerosol particles of terrestrial or marine origin may also serve as warm INPs. These particle types may include pollen, viruses, bacterium or microscopic plant material (Pruppacher et al., 1998; Hoose et al., 2010; Murray et al., 2012). Terrestrial warm INPs can be found in high concentrations near agricultural regions (Tobo et al., 2014), forests (Tobo et al., 2013), and in biomass burning (Petters et al., 2009). Recent studies suggest terrestrial INPs can induce "bioprecipitation feedback" (Huffman et al., 2013; Prenni et al., 2013; Morris et al., 2014; Bigg et al., 2015), whereby rainfall stimulates emission of INPs from some types of terrestrial biota and rainfall efficiency is thereafter increased. Some terrestrial INPs are not often associated with long-range transport, because they are quite large in size (Diehl et al., 2001, 2002; Möhler et al., 2007) and are thought to be efficiently removed from the troposphere in precipitation (Stopelli et al., 2015). Marine INPs are thought to be an important source for the global INP budget (Burrows et al., 2013). Indeed, it has been shown that biological material ejected to the atmosphere in sea-spray may contribute to immersion mode freezing at temperatures as warm as -5 °C (DeMott et al., 2016; Wilson et al., 2015; McCluskey et al., 2018).

ARs often exist near upper tropospheric jet streams and can generate deep clouds whose tops may access airmasses containing long-range transported (LRT) dust or dust/bio INPs. Both Ault et al. (2011) and Creamean et al. (2013) hypothesized that INPs arrived to their storms near cloud top and showed through back-trajectory analysis that the likely sources of these INPs were Asiatic, Arabian and African desert regions. The degree to which terrestrial or marine warm INPs enter AR clouds is less well-established, though good evidence that marine aerosols and terrestrially emitted pollutant aerosols enter the clouds in ARs over California has been provided (Rosenfeld et al., 2008, 2014).



The impact warm INPs have on AR clouds is likewise not established. ARs support a wide variety of clouds, cloud structures and kinematic features that could allow warm INPs to encounter supercooled liquid droplets. Past authors have noted that ARs regularly generate stratiform orographic clouds containing a large amount of supercooled liquid water (Heggli et al., 1983; Heggli and Rauber, 1988), and that AR orographic clouds regularly form seeder-feeder structures (Robichaud and Austin, 1988) wherein falling ice hydrometeors grow rapidly by riming in the warmest supercooled layers (Neiman et al., 2002; White et al., 2003; Creamean et al., 2013). In the seeder-feeder model, the altitude or temperature of warm INP injection to the cloud may lead to differing hydrometeor growth outcomes by changing the relative importance of processes such as riming, ice multiplication, and/or the Wegener-Bergeron-Findeisen process (Pruppacher et al., 1998). Further complicating matters, the type of cloud, depth of cloud and amount of supercooled liquid water may vary considerably during a given AR and could depend upon local topography and short-lived kinematic forcing mechanisms (Kingsmill et al., 2006).

While the authors mentioned above and others have collected INPs in AR clouds and precipitation and found important links between INP source and ARs, additional contrast between local marine and terrestrial and LRT warm INP sources is needed. In addition, coincident analyses of warm INPs with cloud injection temperature and hydrometeor properties is necessary to establish that warm INPs impact AR clouds rather than simply becoming removed by below-cloud precipitation. Hereafter, we will refer to local marine and terrestrial warm INPs as simply "marine" and "terrestrial". LRT will refer to all other warm INPs (see sections 3.1 and 3.3 for details).

For the current study, we examined hourly precipitation samples collected at two Northern CA, USA locations during an AR during 5 - 6 March, 2016. During an extended period of this event, the coastal site - Bodega Marine Laboratory, CA - was directly upwind of the inland site - Cazadero, CA by approximately 35 km. We will demonstrate that the geometry of the flow during this AR and the geography of the two sites create a natural contrast whereby both sites were exposed to marine and long-range transported aerosol sources, but only the inland site was exposed to terrestrial aerosol sources. We will additionally demonstrate that the temperature lapse rates of this storm and partial beam blocking by the coastal mountain range near the measurement sites constrained weather service radar such that retrieved hydrometeor information was indicative of mixed phase clouds in the range -10 °C < $T \leq$ 0°C. We used these unique properties to inform analyses of the amount and activation spectra of ice nuclei in precipitation, cloud macrostructure, cloud hydrometeor phase, kinematic forcing mechanism, cloud-terminating airmass source and transport patterns. These analyses allowed us to address the following questions:

1. What roles do terrestrial, marine and LRT sources have in determining the warm INPs during this AR?

2. What are the transport patterns and cloud injection termperatures for each of these sources?

3. How does meteorology (including bioprecipitation feedback) modulate the source strength and injection temperature and thus the impact of the INP source?

4. When warm INPs are present in precipitation, are cloud microphysics impacted?

The rest of this study will be organized as follows. We present data sources and the study location in section 2. Methodology, including the detection of kinematic forcing regime; Lagrangian dispersion modelling; and radar analyses are presented in





section 3. We will review the atmospheric river event and present our findings in section 4. Finally, we review how our findings address the above questions in section 5.

## 2 Data sources and study locations

### 2.1 Atmospheric river observatory

The coastal atmospheric river observatory (ARO) was developed by the National Oceanic and Atmospheric Administration Earth System Research Laboratory (NOAA-ESRL) to better observe kinematic forcing, cloud and precipitation processes during landfalling ARs. The ARO is comprised of two sites in Northern CA commonly exposed to AR conditions during the winter months. A coastal site at Bodega Marine Laboratory in Bodega Bay, CA (BBY; 15 m MSL; 38.32 °N, 123.07 °W) and a mountain site in Cazadero, CA (CZC; 478 m MSL; 38.61°N, 123.22 °W), together measure nearly coincident weather

conditions during landfalling ARs (White et al., 2013). During the event described herein, both sites had a tipping bucket raingauge, near-surface (10 m) anemometer, GPS receiver capable of estimating integrated water vapor by means of radio occultation, and a vertically oriented radar for vertical sensing of atmospheric properties. BBY had a 449 MHz wind profiling radar and CZC had a S-band precipitation radar (See table 1 for a list of all ARO measurements and their technical references). The CZC S-Band radar was used to determine the echo top height (ETH - see section 3b) during the AR. Neiman et al.

(2002) contains a description of the coastal ARO, and application of the measurements to AR kinematics, cloud properties and precipitation.

### 2.2 Precipitation samples

Precipitation samples were collected hourly from 00 UTC on 5 March to 00 UTC on 7 March, 2016. Precipitation was captured by the Teledyne ISCO model 6712 commercial water samplers, (Teledyne ISCO, Inc., US) connected by tygon tubing to a

300 mL funnel. Precipitation was dispensed into one of twenty-four 350 mL glass jars with hourly collection time interval. Sampling began by manually initiating the program on the sampler at BBY and by triggering from the Teledyne ISCO 674 rain gauge, set to 0.5 mm threshold, at CZC. Two ISCO samplers, programmed to sample sequentially, were placed at each site, enabling a 48 hour continuous collection period. Prior to collection, glass jars were cleaned with acetone, methanol, and ultrapure milli-Q water (18 M$\Omega$ cm$^{-1}$) and peripheral hardware (funnel, tubing, distributor arm, etc.) was rinsed with

milli-Q water. Precipitation samples analyzed in the automated ice spectrometer (section 2d) were separated into 40-mL glass scintillation vials, frozen and stored at -20 °C for approximately 4 months before they were thawed for analysis.

### 2.3 Balloon-borne soundings

Helium balloon-borne GPS-rawinsondes (Vaisaila model RS-41) were launched from BBY at irregular interval varying from 60 to 180 minutes during AR1 and AR2. Each rawinsonde carried a package of meteorological instruments to measure ambient

temperature, humidity, latitude, longitude and altitude. This data was broadcast to a ground-based antenna at BBY during

(c) Author(s) 2018. CC BY 4.0 License.





balloon flight. Two-dimensional horizontal wind was derived automatically from the time-derivative of rawinsonde position. Vaisala model MW41 DIGICORA sounding system software was used to postprocess and archive data from each rawinsonde. The relevant soundings used in analysis of the AR event are listed in Table 2.

### 2.4 Climate Forecast System

NOAA Climate Forecast System (CFS) global short-duration ($t < 6$ hr) forecasts (Saha et al., 2014) were used as three-dimensional atmospheric forcing datasets for FLEXPART (Section 3b). CFS was also used to identify laregescale meteorological features such as ARs and the Pacific upper tropospheric jet stream.

### 2.5 Automated Ice Spectrometer

INP concentrations and freezing activation spectra were determined via the droplet freezing method (Hill et al., 2014) using
the automated ice spectrometer (AIS - Beall et al. (2017)). Precipitation samples were distributed directly in microliter aliquots into a 96-well polypropylene assay plate. Each run consisted of 3 - 5 precipitation samples, along with a milli-Q water sample as control for contamination from the loading. The assay plates were loaded into the AIS, which was cooled until the samples are frozen. Though the homogeneous freezing point of water is -38 °C, freezing of milli-Q samples typically started at -25 to -27 °C, effectively setting the cold limit at which freezing due to INPs in precipitation can be determined. Cumulative
droplet freezing activity spectra, $INP(T)$ (mL$^{-1}$ rainwater), were calculated using the fraction of unfrozen wells $f$ per given temperature interval: $INP(T) = ln(f)/V$, where $V$ is the volume of the sample in each well (Vali, 1971). The fraction of unfrozen wells f was adjusted for contamination by subtracting the number of frozen milli-Q water wells per temperature interval from both the total number of unfrozen wells and the total number of wells of the sample. Warm INP concentration, $INP_{-10}$, is herein defined as the cumulative concentration at $T$ = -10 °C.

A companion set of precipitation samples were heated prior to introduction to the AIS to detect ice-nucleating biological material that is sensitive to heat (Hill et al., 2014). Heated precipitation samples were subjected to heat via immersion in a hot water bath (90 °C) for 20 minutes prior to analysis with the AIS. In analysis presented later, if heated $INP(T)$ decreased compared to un-heated $INP(T)$ drawn from the same precipitation sample for T < -10 °C, we consider a portion of warm INPs from that sample to be bio-INP.

## 3   Methods

### 3.1 Identifying features in radar, soundings and atmospheric model data

To address questions 2 and 3 we sought to identify features of cloud macrostructure (radar brightband and echotop height), kinematic forcing mechanisms (AR low-level jet, coastal barrier jet, polar cold front), large-scale meteorology (AR, Pacific upper tropospheric jet stream) and airmass source regions (terrestrial, marine boundary layers). A short definition of each and
identification methodology using study datasets is to follow:



1. Brightband height (BBH): This quantity is reported by the S-Band (449 MHz) vertically pointing radars at CZC (BBY). See White et al. (2013) for details.

2. Echotop Height (ETH): This quantity is reported by the S-Band radar at CZC. See White et al. (2013) for details.

3. Atmospheric river (AR): AR were identified according to the method of Rutz et al. (2014) using a minimum $IVT$ threshold of $250 \, \mathrm{kg \, m^{-1} \, s^{-1}}$, and a minimum along-vapor transport length of $2000 \, \mathrm{km}$. CFS data were used to identify AR.

4. Pacific upper tropospheric jet stream (UTJ): The UTJ was identified using CFS data when horizontal wind speed exceeded $50 \, \mathrm{m \, s^{-1}}$ between an altitude of 6.5 and $11 \, \mathrm{km \, MSL}$ (hereafter referred to as the UTJ layer). The UTJ layer was defined by visual identification of the UTJ in latitude-vertical cross-sections along the longitudes 135, 150, 165 °W extending from 25 °N to 60 °N during 05 March and 06 March, 2016.

5. AR low-level jet (LLJ): The LLJ was defined as a time-height maximum in terrain-normal water vapor flux occurring below $3 \, \mathrm{km \, MSL}$ (Neiman et al., 2002; Ralph et al., 2005). Terrain-normal water vapor flux was calculated from rawinsondes following the formula $|\boldsymbol{u}|q_v$ , where $q_v$ is the water vapor mixing ratio $(\mathrm{g \, kg^{-1}})$ and $|\boldsymbol{u}|$ is the magnitude of the horizontal wind $(\mathrm{m \, s^{-1}})$ projected along the terrain-normal (upslope) direction for the ARO local terrain (Neiman et al., 2002). A hypothetical wind barb directed along the upslope direction ($\hat{u}$) is depicted in Fig. 1. Rawinsonde observations of two-dimensional wind speed and $q_v$ were temporally interpolated to a constant 60 minute time-series using cubic-spline before water vapor flux calculations were performed.

6. Coastal barrier jet (CBJ): The CBJ was defined as a time-height maximum in along-slope water vapor flux occurring below the local terrain height - (450 m Neiman et al. (2004)). along-slope water vapor flux was calculated similarly to terrain-normal water vapor flux, except the formula is expressed $|\boldsymbol{a}|q_v$, where $|\boldsymbol{a}|$ is the magnitude of the horizontal wind projected to the along-slope direction ($\hat{a}$). See a hypothetical along-slope wind barb depicted in Fig. 1

7. Polar cold front: The polar cold front was identified using rawinsonde data by the directional wind shear in the lowest 5 km of the troposphere. The discontinuity between horizontal wind veering/backing with height (Neiman et al., 1991) is considered to mark the transit of the cold front across the ARO.

8. Marine boundary layer (MBL): For inclusion in the MBL, the geopotential height must have been less than the FLEXPART analyzed planetary boundary layer depth (see section 3b) and the geolocation must have been over the Northeast Pacific Ocean.

9. Terrestrial boundary layer (TBL): The TBL was identified similar to the MBL, except the geolocation must have been over the US state of California.



### 3.2 FLEXPART simulations of backward dispersion from similar cloud injection temperatures

We used the FLEXPART Lagrangian dispersion model (Stohl et al., 2005) to simulate backward dispersion from discrete cloud layers over the ARO. FLEXPART version 9.0.2 was run in serial processor mode on a Unix workstation. The vertical boundaries of cloud layers with similar injection temperature were identified using rawinsondes and the S-Band radar. Mixed phase cloud

layer boundaries were assigned by the geopotential height corresponding to 0 °C and -12 °C, respectively. Cloud top layer upper and lower bounds were assigned by perturbing ETH by +/- 500 m, respectively. . Backward simulation of FLEXPART element position (latitude, longitude and altitude) was performed individually for mixed-phase and cloud top layers over the ARO. Two thousand elements were released per layer for three consecutive hours surrounding significant kinematic features (section 3a). Element position was simulated backward in time for 120 hours prior to release.

It should be noted that the distance separating BBY and CZC is approximately 35 km, and is less than the horizontal resolution of the CFS grid (0.5 degree latitude by 0.5 degree longitude). However, FLEXPART performs several operations designed to resolve motions at less than grid scale (Stohl et al., 2005). We ran FLEXPART simulations for each site separately, but with small exception did not find significant difference in element position or transport patterns. Therefore, unless noted we present only the result of FLEXPART backward simulations ending at CZC.

### 15 3.3 Quantitative analysis to link meteorological features, airmass source and cloud injection temperature

We quantitatively assessed whether air arriving in the mixed-phase or cloud top layers passed through the AR, UTJ, MBL or TBL along its transport path. We calculated the probability of instantaneous element residence, $P_{res}$, in each feature. Each element position was considered an instantaneous sample from a set of elements that would end in the mixed-phase (cloud top) layer over the ARO. The quantity $P_{res}$ was calculated as the fraction of the set of positions that could be assigned to the

features. This is expressed as $P_{res} = n_{res}/n_{rel}$, where $n_{res}$ is the number element positions residing in the desired feature, and $n_{rel}$ is the total number released from the given layer above the ARO. CFS horizontal wind speed, relative humidity, $IVT$, land/ocean mask, and FLEXPART boundary layer depth were linearly interpolated to element position to assess whether a given element resides in a feature. AR residence was assigned to a FLEXPART element location if the atmospheric river definition from section 3b was instantaneously met and the relative humidity exceeded 85%. UTJ, MBL and TBL residence

were assigned based upon the definitions in section 3a. It was found that the number elements satisfying the TBL condition was always zero if the time to arrival (TOA) at the ARO was more than 3 hours (e.g. the flow was sufficiently strong that the ARO was well-ventilated). Therefore, the quantity $P_{TBL}$ was calculated from a truncated set of elements: $P_{TBL} = n_{res}/n_{TOA \leq 3}$, where $n_{TOA \leq 3}$ is the number of elements released less than 3 hours ago from the given layer.

   In summary, the methods from sections 3a-c have been used to link INP source regions to clouds over BBY and CZC via

transport in largescale meteorological features such as the AR and UTJ by means of FLEXPART simulations. Performing separate FLEXPART simulations for cloud layers discretized by injection temperature and for time periods discretized by kinematic forcing further addressed the questions related to cloud injection temperature and modulation by meteorology. It is important to clarify that we cannot perform a complete budget for INP source regions. For example, we can identify proxy




regions for local INP sources using the terrestrial and marine boundary layers, but these methods cannot capture all possible LRT source regions. Thus, we must in part make inferences about source after rejecting alternate hypotheses if the mechanisms examined are not supportive.

### 3.4 Analysis of KDAX weather service radar retrievals

The KDAX weather service radar (Heiss et al., 1990) located in Sacramento, CA was used to evaluate hydrometeor phase and precipitation intensity in a shallow mixed-phase cloud layer over BBY and CZC. The location of KDAX relative to BBY and CZC is shown in Fig. 2a. During each azimuthal scan, the lowest beam elevation (0.51 degree) from KDAX is partially blocked by the coastal mountain range. The result of the beam blockage is that signal is only returned from a narrow vertical slice of the scan above BBY (CZC). Figure 2b depicts the blocked and unblocked portions of the beam and the portion of the

atmosphere that is sampled above both sites during each scan (red trapezoid in Fig. 2b). The upper and lower altitudes of the KDAX unblocked layer are 2850 and 3650 m, respectively. During this storm, rawinsondes measured the temperature range corresponding to the upper ($T_{KDAX}^{top}$) and lower ($T_{KDAX}^{bot}$) limits of the shared KDAX unblocked layer (Table 2). Hydrometeors sensed by KDAX in the unblocked layer over BBY and CZC were in the temperature range 0.8 °C to -9.2 °C. Therefore, information retrieved from the KDAX unblocked layer such as hydrometeor phase and radar reflectivity were indicative of

warm mixed-phase clouds during the storm.

   We investigated whether the likelihood of detecting ice phase precipitation (hereafter the category "snow") was independent of ARO site. To do so, we cultivated a sample of KDAX hydrometeor classification retrievals (Park et al., 2009) for the azimuth and range gates approximately corresponding to CZC and BBY. The weather service radar hydrometeor retrieval contains 11 classifications: biological (animals, not particles), clutter, ice, dry snow, wet snow, light rain, heavy rain, big drops, graupel,

hail, and unknown. We grouped the classifications ice, dry snow, wet snow and graupel into our snow category. All other classifications beside unknown were categorized as "not-snow".

   In the results section, we present the likelihood of observing snow category hydrometeors during precipitation in the KDAX unblocked layer for each site ($P_{snow}^{BBY}$, $P_{snow}^{CZC}$). We sought to preserve the mixed phase temperature range as found by the soundings in Table 2. To this end, we retained only one range gate, nearest to the CZC site, from the CZC azimuth. We retained

the 45 range gates from the BBY azimuth that complete the red trapezoid in Fig. 2b. The KDAX radial resolution is 250 m, thus the BBY azimuth retrievals correspond to the unblocked layer over BBY and along a great circle toward KDAX extending 11.5 km. To test the association between $P_{snow}^{BBY}$ and $P_{snow}^{CZC}$, The binary categorical data from BBY and CZC range gates and azimuths were used to perform a Chi-Square independence test. We verified that the data passed the rule of thumb for minimum expected populations and Yates' correction (Haviland, 1990).

We do not possess independent observations of temperature in the KDAX unblocked layer over each site. Instead, we assume that the temperatures $T_{KDAX}^{top}$ ($T_{KDAX}^{bot}$) are equivalently representative of both sites. Each rawinsonde's ground location was tracked to an altitude of 3650 m. The mean ground location in the height range of the KDAX unblocked beam layer varied by sounding but was nearly equidistant from both sites at a distance approximately 19.44 (26.48) km to BBY (CZC). We note that local effects related to airflow over a mountain barrier (Minder et al., 2011) could preferentially cool the lower troposphere





above CZC more than above BBY. If this effect is strong, the unblocked beam layer above CZC could contain cooler air than it does over BBY. Following the methodology of Minder et al. (2011), we performed semi-idealized simulations of flow over a 2-dimensional hill of approximate height (500 m) and half-width (10 km) of the mountain ridge at CZC using rawinsondes from this study as the upstream boundary condition. Simulated temperatures above the CZC proxy mountain were not cooler

than those above the BBY proxy coast by more than 0.25 °C.

## 4  Results

### 4.1  Overview of atmospheric river event

Two ARs impacted Northern California during 4-6 March, 2016. AR conditions (Ralph et al., 2013) lasted 39 hours from 15 UTC on 4 March, 2016 to 06 UTC on 6 March, 2016. This period incorporated both AR events, as there was not a clear break

in AR conditions. Utilizing integrated vapor transport (IVT) from CFS and the method of Rutz et al. (2014), we mark the end of the first AR (hereafter "AR1") at 15 UTC on 5 March, 2016. The second AR (hereafter "AR2") was the stronger of the two by measures of $IVT$ and storm-total precipitation. Measurements at the ARO show that $IVT$ reached a peak value of 956 $\mathrm{kg\,m^{-1}\,s^{-1}}$ near 02 UTC on 6 March (see Table 2). Total precipitation at CZC during AR2 reached 72 mm, placing this event in the top 20% of all events published in Ralph et al. (2013). Moderate AR conditions, defined by $IVT \geq 500\ \mathrm{kg\,m^{-1}\,s^{-1}}$

Martin et al. (2018) or greater, were observed over the ARO for nearly 11 hours (Table 2). Wind speed at 10 m AGL reached a maximum value of 21.7 $\mathrm{m\,s^{-1}}$ near 03 UTC on 6 March. Electric power from the local utility grid was lost at BBY shortly thereafter. A sounding indicated that the cold front on the poleward side of this storm system transited BBY near this time (see also Fig. 5b).

### 4.2  Synoptic scale meteorology

Figure 3 displays the synoptic-scale meteorological conditions over the Northeast Pacific Ocean every 6 hours beginning (ending) before (after) AR2 arrived at (departed) the ARO. Pressure reduced to mean sea-level ($SLP$ - hPa) depicts an extratropical cyclone located near 47 °N; -146 °W at 12 UTC on 5 March. Two distinct troughs, likely associated with fronts are visible in the $SLP$ analysis extending to the east and southeast, respectively, of the cyclone. The troughs and their baroclinic zones support AR1 and AR2, shown in Fig. 3a by the $IVT$ colorfill. Fig. 3 also shows the location of the UTJ using jet layer (see

section 3a) isotachs ($\mathrm{m\,s^{-1}}$). At 12 UTC on 5 March, the upper tropospheric jet is zonal, located along 32 - 34 °N and extends westward from an exit region near 32 °N; -135 °W across the international dateline.

As the event progressed, the extratropical cyclone moved eastward. The cyclone center became located near 49 °N; -133 °W by 06 UTC on 6 March, 2016. The troughs associated with AR1 and AR2 rotated counterclockwise around the extratropical cyclone as they moved eastward. As AR1 dissipated near 18 UTC on 5 March, its trough weakened and thereafter disappears

from the figure. The trough associated with AR2 continued to strengthen through 00 UTC on 6 March, becoming meridionally oriented. $SLP$ contours sharply kink upwind of AR2, indicative of a well-developed polar cold front. The upper-tropospheric





jet remained zonal along or near 32 - 34 °N as the event progressed, while the jet exit region moved eastward toward 31 °N; -125 °W at 06 UTC on 6 March. The movement of the cyclone, troughs and upper-tropospheric jet caused AR1 to move inland and away from the ARO. From this time, AR2 intensified, changed in orientation from southwesterly to southerly, and did not weaken significantly until after it passed the ARO just before 06 UTC on 6 March.

The largescale meteorology surrounding AR2 provide ideal conditions for the study of warm INP sources for two reasons. First, the UTJ remains strong, zonal, extended and quasi-stationary throughout the event. This UTJ represents a clear mechanism for long-range transport of warm INPs from sources beyond the Northeast Pacific Ocean. Second, AR2 transited with a well-defined polar cold front and similar strength AR in this region often contain LLJ and CBJ. As we will see, AR2 indeed contained both these kinematic forcing mechanisms. The remainder of this study will focus on AR2.

**4.3    Warm INPs, rainfall and cloud macrostructure at the ARO**

Figure 4a shows the timeseries of $INP_{-10}$ (box-and-whiskers) and accumulated precipitation (solid lines) during the event at BBY (CZC). Note that $INP_{-10}$ at CZC is consistently between 0.25 and 3 $\mathrm{mL}^{-1}$ before 21 UTC on 5 March and between 3 and 15 $\mathrm{mL}^{-1}$ thereafter. $INP_{-10}$ at BBY only occasionally neared 2 $\mathrm{mL}^{-1}$. The effect is that $INP_{-10}$ at CZC was at least an order of magnitude higher than that at BBY with rare exception. The only AR2 samples for which the AIS registered nonzero

$INP_{-10}$ at BBY occurred between 22 - 23 UTC on 5 March and near 5 UTC on 6 March. The sample collected at BBY at 22 UTC contained $INP_{-10} = 2.67$ $\mathrm{mL}^{-1}$, the highest at BBY during AR2. The heaviest rainfall occurred between 21 UTC 5 March and 3 UTC 6 March at both sites. During AR2 the accumulated rainfall at CZC was approximately double the amount at BBY.

The S-Band radar derivation of $ETH$, and $BBH$ (km MSL) are displayed in Fig. 4b. Also shown is the relative humidity at

5 km MSL ($RH_{5km}$ - %). $BBH$, the altitude of the hydrometeor melting level stayed near 2 km from 15 UTC 5 March to 00 UTC on 6 March. $BBH$ slowly rose beginning 00 UTC on 6 March and briefly spiked to an altitude nearly 3 km near 2 UTC on 6 March. Thereafter $BBH$ descended rapidly to 1.5 km MSL after the transit of the cold front. $ETH$, the radar estimated height of cloud top, was more variable during the event. S-Band retrievals are intermittently missing between 15 UTC and 21 UTC on 5 March, suggesting a lack of precipitating hydrometeors during some of this period. For non-missing Early AR

retrievals, the median value of $ETH$ was near 5 km MSL before 21 UTC on 5 March. $ETH$ rose sharply after 21 UTC on 5 March, reaching an event maximum value just over 8 km MSL. After 23 UTC on 5 March, $ETH$ fell to a minimum value of approximately 4 km MSL at 02 UTC on March 6. This time corresponds to the maximum measured $IVT$ (Table 2). $ETH$ rose again near cold front passage, passing 7 km MSL. After 5 UTC on 6 March, $ETH$ fell precipitously. After 6 UTC, S-Band retrievals of $ETH$ and $BBH$ disappeared.

After 18 UTC on March 5, ETH and $RH_{5km}$ are qualitatively well correlated. Echo top heights rose (fell) in the range 4 km MSL to 8 km MSL as $RH_{5km}$ rose (fell). This suggests that the availability of moisture was a factor controlling the presence of upper cloud layers during this event. It is noteworthy that the strongest $IVT$ and heaviest rainfall occurred when mid-levels were dry, cloud tops were lower, and $INP_{-10}$ were absent at BBY. We will explore whether warm INPs in BBY precipitation is related to cloud top altitude in section 4f. Background coloration and labels ("Early AR", "Barrier Jet", "Peak





AR", and "Post CF") refer to periods of dominant kinematic forcing and their dominant kinematic feature. These kinematic periods will be introduced and described in the next section.

While both sites experienced very similar weather conditions during AR2, warm INPs were much more prevalent in precipitation at CZC than at BBY. The enhancement in $INP_{-10}$ (Fig. 4a) was more than a factor of 10 during most of the storm.
While $INP_{-10}$ remained elevated throughout the latter three periods in CZC precipitation, $INP_{-10}$ presence in BBY precipitation was ephemeral. These two findings suggest that the two sites were exposed to different warm INP sources, experienced different cloud injection mechanisms, or both.

## 4.4 Kinematics and periods of AR2

Time-vertical meteograms of along-slope and upslope vapor flux $(\mathrm{g\,kg^{-1}\,s^{-1}})$ over the ARO are shown in Fig. 5a. Along-slope
(upslope) vapor flux is here used to indicate transport of water vapor consistent with a coastal barrier jet (AR low-level jet) - see section 3a. CBJ vapor transport reached its maximum between 21 and 23 UTC on 5 March. Maximum values in along-slope vapor transport were located between the surface and $400\,\mathrm{m\,MSL}$. The LLJ vapor transport maxima occured later, between 23 UTC on 5 March and 01 UTC on 6 March. the LLJ vertical maxima was located above the height of the coastal mountains, near $750\,\mathrm{m\,MSL}$. This spatio-temporal evolution of the CBJ and LLJ is consistent with previous studies. In particular, Neiman
et al. (2004) found that the barrier jet typically forms before the arrival of maximum vapor transport, in response to blocking of the flow by local topography. Kingsmill et al. (2013) described the AR low-level jet as forced upward over the top of an antecedent barrier jet. Ralph et al. (2005) found that the low-level jet is responsible for the majority of the horizontal vapor flux in AR, and that the typical vertical location of the low-level jet is near $1\,\mathrm{km\,MSL}$.

Fig 5b shows horizontal wind $(\mathrm{m\,s^{-1}})$ vectors from balloon-borne radiosondes. The top axis indicates the time of soundings
measuring $IVT$ values of (514, 736, and 956) $\mathrm{kg\,m^{-1}\,s^{-1}}$, respectively. Also indicated on the top axis is the time of sounding indicating the transit of the cold front. Wind barbs back with height in the lowest 5 km of the troposphere for this and following sondes, further indicating the cold front has passed. The strength of each the coastal barrier jet, the low-level jet and the cold front, along with their interchange in a short period of time suggests that the kinematic forcing for orographic clouds during this AR may have changed rapidly several times. We will hereafter use the break between AR1 and AR2, the dominance in
vapor flux by the CBJ (LLJ), and the transit of the cold front to segment AR2 into four kinematic periods (see Table 3).

## 4.5 Droplet freezing spectra at BBY and CZC and their response to heat treatment

Figure 6a,c show the droplet freezing activation spectra, $INP(T)$, as measured by the AIS from precipitation samples at BBY and CZC, respectively. Vertical lines at -10 °C are provided so that the number of warm INPs is visually enhanced. In CZC samples, significant freezing events occurred for $T >$ -10 °C in all periods. Concentrations from CZC in the temperature
range -15 °C $< T \le$ -5 °C are consistent with precipitation samples containing terrestrial bio-INPs as reported in Petters and Wright (2015). In the Barrier Jet and Peak AR periods, freezing events were detected at temperatures as warm as -5 °C. In agreement with Fig. 4a, few BBY samples from the Barrier Jet period and one sample from the Post CF period similarly contained material that froze at $T >$ -10 °C. As time passed during AR2, the maximum $INP(T)$ and $INP_{-10}$ both increased





in precipitation collected at CZC. Concentrations were greater during the Peak AR period than during Barrier Jet; and Barrier Jet concentrations were greater, in turn, than during Early AR. Rainfall also accumulated over time, with the sharpest increase in rain rate between the Early AR and Barrier Jet periods. We do not have sufficient analysis to confidently ascribe the increasing trend in $INP_{-10}$ to bioprecipitation feedback (Huffman et al., 2013; Prenni et al., 2013; Morris et al., 2014; Bigg et al., 2015).

It is also noteworthy that no such trend in $INP_{-10}$ is apparent in BBY precipitation. We might expect rain falling on the forest savannas and and pasture lands (see Fig. 1b) between CZC and the Pacific Ocean during the Early-AR periods to progressively stimulate emission of terrestrial warm INPs. An increase in warm INP source strength over time through a process similar to bioprecipitation feedback may explain some of the temporal trend in $INP_{-10}$ at CZC. BBY, however, is not downwind of any sources that are known to respond to precipitation in this way.

Further difference in INPs between BBY and CZC is found in the shapes of the freezing spectra. The freezing spectra for $T <$ -10 °C at BBY (Fig. 6a) are log-linear and negatively sloped with temperature. This agrees with models predicting immersion mode freezing of dust published by DeMott et al. (2010) and Niemand et al. (2012). Fig. 6c, by contrast, shows freezing spectra from CZC that must be modeled by more exotic functions of temperature. This is consistent with immersion freezing of bio-INPs (Murray et al., 2012; Tobo et al., 2013, 2014; Petters and Wright, 2015). Thus, it is likely that biological

material contributed significantly to INP concentrations for $T <$ -10 °C at CZC, but not at BBY.

Figures 6b,d show the fractional change in $INP(T)$ after precipitation samples from BBY and CZC, respectively, were heated (see section 2d). This is expressed as $\Delta INP(T)/INP(T)$, where $\Delta INP(T)$ is the concentration from the unheated sample minus the concentration at matching temperature from the heated sample. Heating the precipitation samples prior to measuring their freezing activation de-natures biological material that would otherwise have supported ice nucleation (Hill

et al., 2014, 2016). It may also cause insoluble inorganic material to break apart. In some cases, this fracturing of insoluble material can lead to increases in $INP(T)$ (McCluskey et al., 2018). For $T <$ -15 °C, the combination of these effects may lead to a mixture of positive and negative $\Delta INP(T)$. Additionally, heat treatment may completely nullify the ability of some bio-INPs to support freezing but may not render other types (e.g. cellular fragments) freezing inactive.

At both sites, heating nullified most freezing for $T >$ -10 °C. The exception is for samples during the Peak AR period at

CZC. Some CZC Peak AR samples partially, but not completely, lost their freezing activity for $T >$ -10 °C after heating. The difference spectra for both sites support the conclusion that biological material is serving as warm INPs. The issue of mixed trend in $\Delta INP(T)$ for $T <$ -15 °C is apparent in samples from both sites. $INP(T)$ increased after heating in 23% (11%) of samples collected at BBY (CZC), respectively.

### 4.6  Qualitative transport patterns and their association with warm INPs in precipitation

The location and altitude of FLEXPART elements released in the mixed-phase and cloud-top layers for each of the four periods are displayed in Figure 7. Of note for understanding LRT warm INPs during the AR, Fig. 7b,d display the element position for releases made during the Barrier Jet and Post CF periods. During these periods, many elements ending in the cloud top layer travelled along the upper tropospheric jet stream. Recall from section 3a that the jet stream is located between altitudes of 6.5 and 11 km MSL, therefore yellows, oranges and reds in figure 7 indicate appropriate jet altitudes. By contrast, element positions





for cloud-top releases during Early AR (Fig. 7a) and Peak AR (Fig. 7c) periods do not visually show transport influence from the jet stream. The difference in degree of jet stream influence between the three pre-cold frontal periods likely comes from cloud-top layer altitude (Table 2). Elements ending in the cloud-top layer during the Post-CF period likewise appear to have travelled along the Pacific upper tropospheric jet even though cloud tops were lower during much of this period. Subsidence in the post cold-frontal airmass may have linked the high-altitude UTJ and relatively lower cloud tops during the Post CF period. The Barrier-Jet and Post-CF periods were the only periods during which warm INPs were detected in precipitation collected at BBY (Fig. 6a). Figures 6a and 7 together suggest some long-range transported warm INPs may have arrived to the AR cloud tops by travelling across the Pacific Ocean on the upper tropospheric jet stream. This result is in broad agreement with findings in Ault et al. (2011) and Creamean et al. (2013). From Fig. 7 it is also apparent that elements nearing the ARO primarily travelled along the AR during the final hours of their flight. The AR played a smaller role in transport to the cloud layers during the Post CF period, when lower tropospheric airmasses arrived to the ARO from the cold-sector, or from the west of the cold front and AR, just glancing the AR upon arrival.

### 4.7 Quantitative relationships between airmass source, transport mechanism and cloud injection temperature

Table 4 presents the probability of element residence (section 3c) in the UTJ, AR, MBL and TBL. From Table 4, one can verify many of the broad qualitative findings from Figure 7. Namely, elements were much more likely to arrive in the cloud-top layer after travelling in the UTJ during the Barrier Jet and Post CF periods; airmasses arriving in the mixed-phase layer had the largest marine boundary layer influence during the Barrier Jet and Peak AR periods; and the probability that an element passed through the AR before arriving in the clouds above CZC is smallest for the Post CF period.

Table 4 also offers insight to which periods were most likely to have terrestrial boundary layer air drawn into the mixed-phase cloud layer. The probability that an element both travelled through the terrestrial boundary layer and ended in either cloud layer during the Early AR period is zero. Likewise, there is zero probability that elements travelled through the terrestrial boundary layer and entered the cloud-top layer during any period. The probability that an element travelled through the terrestrial boundary layer and ended in the mixed phase cloud layer above CZC during the Barrier Jet, Peak AR and Post CF periods is 0.062, 0.083, and 0.044, respectively. Note that all elements arrived at CZC in both layers from the west or southwest (offshore) during all periods, and thus had a very short trip over or through terrestrial boundary layers. These directions of travel were the same for FLEXPART simulations over BBY (not shown). The location of BBY directly on the coastline thus yields $P_{TBL} = 0$ for all layers and all periods. BBY clouds were never downwind of a nearby landmass during AR2. We interpret the $P_{TBL}$ results to mean that approximately 4% to 8% of the air arriving in the mixed-phase clouds over CZC also spent time in the boundary layer over nearby land surfaces. If the local terrestrial biomes were a source of warm INPs, mixed phase clouds were able to entrain warm INPs into layers that could support heterogeneous freezing. Note that the concentration of $INP_{-10}$ at CZC increased markedly from the Early AR period to the other periods considered (Fig. 4a), following the trend in increasing $P_{TBL}$. As discussed (section 4e), an increase in terrestrial warm INP source strength may also explain the increase in $INP_{-10}$ over time during AR2. We are not able to disentangle the two effects here.





We can now address questions related to warm INP source and injection mechanism. Both sites were downwind of marine particle sources for the entire storm and the cloud layers above each site received significant contributions from the marine boundary layer during all storm periods. Only CZC precipitation contained warm INPs during all periods. The only persistent difference in airmass influence between the cloud layers over the two sites was that inflowing air to CZC passed through

the terrestrial boundary layer before arriving. Thus, we conclude that the warm INPs present in CZC precipitation are predominantly terrestrial in origin, and that terrestrial warm INPs are not found in BBY precipitation. There is no mechanistic explanation for the simultaneous presence (lack) of warm INPs at CZC (BBY) if the warm INP source is marine. LRT warm INPs were ephemerally present, likely at both sites. LRT warm INPs were injected at cloud top and their transport and injection were highly modulated by large-scale meteorology (e.g. the UTJ), kinematic forcing mechanism and the availability of mid-

tropospheric moisture. The cloud top temperature, and thus injection temperature for LRT warm INPs, may vary considerably (See $ETH$ in Barrier Jet, Post CF periods in Fig. 4b).

Table 4 demonstrates that a transport pathway existed for terrestrial boundary layer air, potentially containing terrestrial INPs, to become injected to mixed phase clouds. The activity of this pathway ($P_{TBL}$) was modulated by kinematic forcing regime. For example, it was inactive during the Early AR period. Terrestrial boundary layer air travelled this pathway through

a precipitating cloud base, and were thus at risk for scavenging before they reached subfreezing temperatures. We have yet to demonstrate that terrestrial boundary layer air arrived in mixed-phase clouds while retaining their warm INPs or that those INPs impacted mixed phase cloud hydrometeors. We will investigate those questions next.

### 4.8 Impact of warm INPs on mixed-phase cloud microphysics

Figure 8 displays the timeseries of $P_{snow}^{BBY}$ for each scan. The all-storm value of $P_{snow}^{CZC}$ is displayed as a horizontal reference

line. For the majority of AR2, $P_{snow}^{BBY}$ was much less than the storm-mean $P_{snow}^{CZC}$. The likelihood that KDAX observed snow in the unblocked layer above CZC during the storm is $P_{snow}^{CZC} = 0.615$. The same likelihood over BBY is $P_{snow}^{BBY} = 0.165$. A two-category, two-site Chi-square independence test was performed using all available hydrometeor class retrievals from each site. The null hypothesis, that the likelihood of observing snow is independent of site, is rejected with $P = 4.3 \times 10^{-38}$. This result is insensitive to Yates' correction. By visual inspection of Fig. 8 and by the result of the Chi-square independence test,

snow hydrometeors were more likely at equivalent temperatures over CZC than over BBY. As we have seen, warm INPs were also consistently more numerous, by as much as a factor of 10, in CZC precipitation. We can thus hypothesize that terrestrial warm INPs become injected into mixed-phase clouds over CZC and impact cloud hydrometeor populations through in-situ ice-phase microphysics. Also of note in Fig. 8, $P_{snow}^{BBY}$ did not increase during the Barrier Jet period, though Barrier Jet period precipitation samples from BBY contained higher $INP_{-10}$. It is possible that warm INPs over BBY were only injected through

cloud top at colder temperatures, supporting the activation of other INP sources. If so, LRT warm INPs may have minimally impacted the presence of snow in the mixed-phase layer. This explanation is consistent with the LRT source and injection mechanisms found for BBY in prior analyses.

Because we cannot directly measure the impact of warm INPs on $P_{snow}^{BBY}$ ($P_{snow}^{CZC}$), we must attempt to exclude the possibility of alternate processes explaining the difference in $P_{snow}^{BBY}$ ($P_{snow}^{CZC}$). The first alternate hypothesis we sought to exclude is that



any difference can be explained by differences in the temperature of the unblocked beam layer. We can exclude this alternate hypothesis by noting that the unblocked radar gates sampled to create Fig. 8 and the Chi-square independence test represented the same temperatures over both sites by design (section 3d).

The second alternate hypothesis we address is that any difference in $P_{snow}^{BBY}$ ($P_{snow}^{CZC}$) could be caused by a difference in the rate of snow hydrometeors falling from above the KDAX unblocked layer. To address this possibility, we conducted analysis of the reflectivity in the KDAX unblocked layer over each site. For this analysis only, we relaxed the constraint on temperatures above each site in favor of also retaining 45 gates from the CZC azimuth. Radar reflectivity is closely related to the precipitation rate, thus a strong association between the KDAX unblocked reflectivity and $P_{snow}^{BBY}$ ($P_{snow}^{CZC}$) is considered to indicate that snow category hydrometeors are primarily falling from higher and colder layers. Radar power is also returned more strongly for liquid hydrometeors than for ice hydrometeors. Therefore, in the absence of any relationship between strength of precipitation rate and likelihood of snow, we should expect a weak negative relationship between reflectivity and the likelihood of snow in the unblocked layer. We also note that inter-site comparisons of reflectivity are not appropriate, since the degree of beam blockage is different over each site and we do not perform any correction to retrieved beam power based on the blockage geometry (e.g. Qi et al., 2014).

The relationship between $P_{snow}^{BBY}$ ($P_{snow}^{CZC}$) and mean unblocked layer reflectivity for all scans is shown in Figure 9. Note there is little to no correlation between mean reflectivity and $P_{snow}$ for either site. $R^2$ is 0.004 (0.006) for BBY (CZC). Least squares yields a very weak positive slope, in $\mathrm{dBZ}^{-1}$, for each site. We thus conclude that the precipitation rate had very little effect on the chance of observing snow in the unblocked layer over both sites. It is possible that a weak relationship between chance of observing snow and precipitation rate did exist during the Peak AR period. Figure 9b shows the most clear evidence of this. Markers in Fig. 9 are colored by their period. The red markers, indicating the Peak AR period, display a slight upward trend in $P_{snow}^{CZC}$ with mean reflectivity. The precipitation accumulation was also greatest during the Peak AR period, so it is possible that snow hydrometeors were falling from colder cloud layers at a greater rate during this period. To exclude the possibility that the Peak AR period skewed the result of the above analysis, we re-computed the chi-square independence test while excluding all retrievals during the Peak AR period. The result did not change. The null hypothesis is again rejected, with $P = 1.1 \text{ x } 10^{-48}$.

## 5 Summary

In this study, we examined the freezing spectra of time-resolved rainfall samples from two Northern CA sites, one coastal (BBY) and one inland (CZC), during an atmospheric river. We compared these spectra and their warm INP concentration ($INP_{-10}$) across sites and across periods categorized by varying kinematic forcing, cloud macrostructure, aerosol source region and transport mechanisms. These analyses were performed to address the following questions. What roles do terrestrial, marine and LRT aerosols play in determining the warm INPs during this AR? What are the transport and cloud injection mechanisms for each of these sources? How does meteorology (including bioprecipitation feedback) modulate the source





strength and injection mechanism and thus the impact of the INP source? When warm INPs are present in precipitation, are cloud microphysics impacted?

In summary, we found

1. Using the AIS, that terrestrial warm INPs are abundant in precipitation at the inland site. It is possible that bioprecipitation
feedback contributes to terrestrial warm INP source for the inland site.

2. Through quantitative analysis of FLEXPART element residence times, that even though a large number of cloud-terminating trajectories passed through the marine boundary layer, we do not see evidence of marine warm INPs at either site during this storm.

3. Through similar analysis, that long-range transported warm INPs may additionally be present in precipitation at both
sites, but only when meteorological patterns, kinematic forcing and cloud macrostructure enable cloud tops to access high altitude transported airmasses.

4. Using the analysis of FLEXPART residence times and radar hydrometeor classifications, we found evidence that terrestrial warm INPs impacted precipitating hydrometeors in mixed phase clouds during this storm.

The first and second findings come from the unique flow geometry and geography of the precipitation collection sites during
AR2. Both sites are downwind of marine particle sources for the entire storm and the cloud layers above each site receive significant airmass contribution from the marine boundary layer during all storm periods (Table 4). However, only the inland site shows warm INPs in precipitation during all periods (Fig 4a and Fig 6a,c). The only difference in airmass influence between the cloud layers over the two sites is that inflowing air to mixed phase clouds over the inland site (CZC) passes through the terrestrial boundary layer before arriving (Table 4). When warm INPs are present in coastal site precipitation, their presence
can be explained mechanistically by transport patterns and cloud macrostructure favorable for LRT aerosols to become injected at cloud top. Conversely, we cannot provide an alternate hypothesis for ephemeral injection of marine warm INPs into coastal site clouds. Here we must note that understanding of marine INP emission processes and activation temperatures is incomplete. It is possible that suppressed emission of marine warm INPs in nearby source regions or offshore removal led to the absence of detectable marine warm INPs during this storm but that marine INPs may be important for other ARs.

The third finding is supported by the ephemeral presence of warm INPs at the coastal site (BBY) during the Barrier Jet and Post CF periods (Fig. 4a). Analysis of FLEXPART elements ending at cloud top found that elements were much more likely to travel through an elongated zonal Pacific jet stream during these periods than during any other (Fig. 7, Table 4).

The fourth finding is supported by two parts. In the first, we investigated whether a mechanism exists to inject terrestrial warm INPs to mixed-phase clouds over the inland site. Analysis of FLEXPART elements arriving to mixed-phase clouds
(Table 4) suggest a small but non-zero probability that terrestrial boundary layer airmasses can become injected to mixed phase clouds. If some terrestrial boundary layer aerosols are also warm INPs, there is a mechanism for some of these particles to reach cloud temperatures where they may stimulate freezing of supercooled drops. Analysis of the KDAX radar hydrometeor retrievals (Fig. 8) further shows that the precipitating hydrometeor phase in clouds with -10 °C $< T \leq$ 0 °C is significantly





different at CZC than at BBY, with a higher probability of snow hydrometeors over CZC. We rejected the alternate hypothesis that snow hydrometeors were more numerous because ice fell from colder layers (Fig. 9). Therefore, we must conclude that in-situ microphysics is making more ice over CZC than over BBY.

As we have seen in multiple analyses presented herein, the role of meteorology in modulating warm INP source, transport and cloud injection mechanism is complex. It depends upon large-scale weather features, kinematic forcing mechanisms such as barrier and low-level jets, and the availability of moisture near cloud top. These are just the processes that determine the warm INPs in the single AR studied herein. ARs as important mechanisms for the removal of trace atmospheric constituents of remote origin and the impact of terrestrial and marine warm INPs on mixed-phase clouds and precipitation are topics deserving further study. Finally, this study demonstrated that polarimetric precipitation radar can be a useful tool to study cloud microphysics given well-constrained conditions. Future studies into the impact of aerosols on cloud microphysics may benefit from targeted polarimetric radar observations conducted in tandem with tropospheric soundings and laboratory analysis of cloud and precipitation material. It is certainly possible to enhance the analysis methods herein and deploy similar methods for multiple storms so that these or future findings may be generalized to other regions or other weather scenarios.

*Code and data availability.* Datasets and code used to create analyses supporting this study are hosted within the UC San Diego Library Digital Collections. https://doi.org/10.6075/J05X274R

*Acknowledgements.* The authors would like to acknowledge the UC Davis Bodega Marine Laboratory in Bodega Bay, CA for providing space for sample collection, laboratory work and housing while the field phase of this study was completed. Many thanks to Drs. Paul DeMott and Thomas Hill of Colorado State University for helpful guidance regarding AIS analyses. National Science Foundation Grants AGS-145147, AGS-1632913, and US Army Corps of Engineers Grant W912HZ-15-2-0019 provided funding for this work. J. Creamean was supported through funding from NOAA Physical Sciences Division.



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



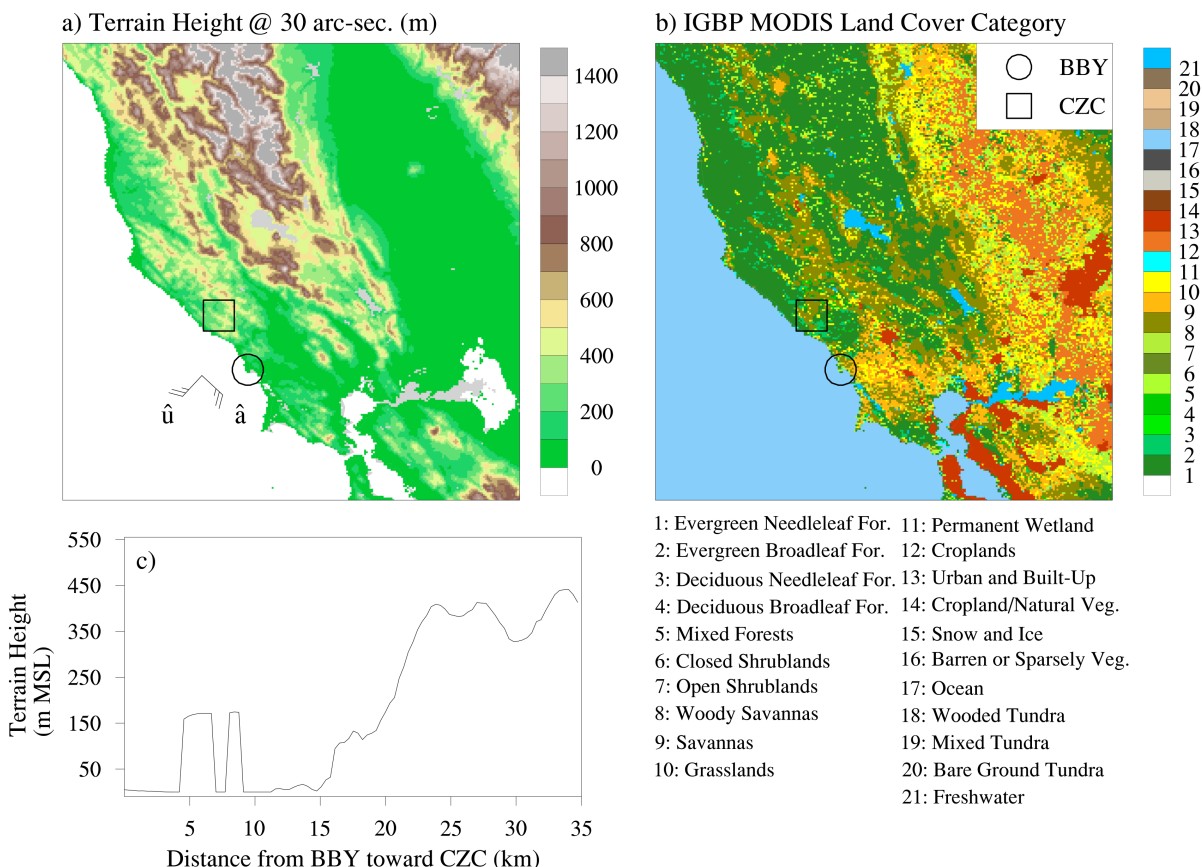

**Figure 1.** a) Plan view of regional terrain height (m - colorfill) from USGS 30 arc-second digital elevation map. Annotations are centered on BBY (circle) and CZC (square) and depict theoretical wind barbs aligned with the upslope (û) and along-slope directions (û). (b) As in a, except the dominant category from the IGBP-MODIS landuse database is depicted (colorfill - see legend for category name). c) Transect of terrain height (m MSL) along a great circle path from BBY to CZC.





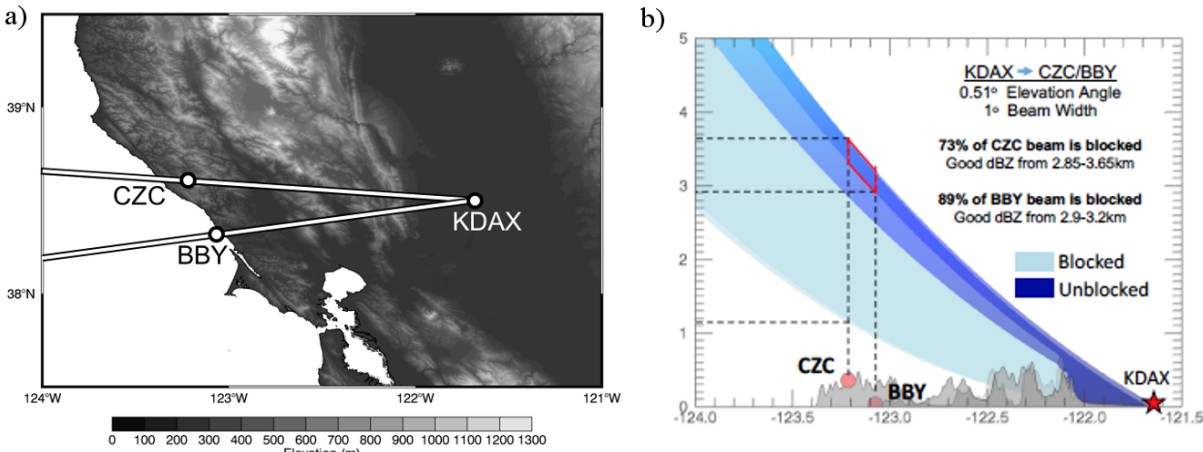

**Figure 2.** a) Plan view of region surrounding the study area with KDAX, BBY and CZC labelled. Beams show path of radar from KDAX to each site (BBY, CZC). White shading indicates relative terrain height (m MSL). b) Height vs. longitude cross-section with KDAX 0.51 degree elevation scan beam blocked (light blue), unblocked over CZC (medium blue) and unblocked over BBY (dark blue) layers. Red trapezoid indicates the volume from BBY azimuths that are unblocked and share the altitudes of the CZC unblocked layer. Location of BBY (CZC) indicated by red dot on ordinate. Terrain profiles along BBY (CZC) azimuths also indicated in gray shading.





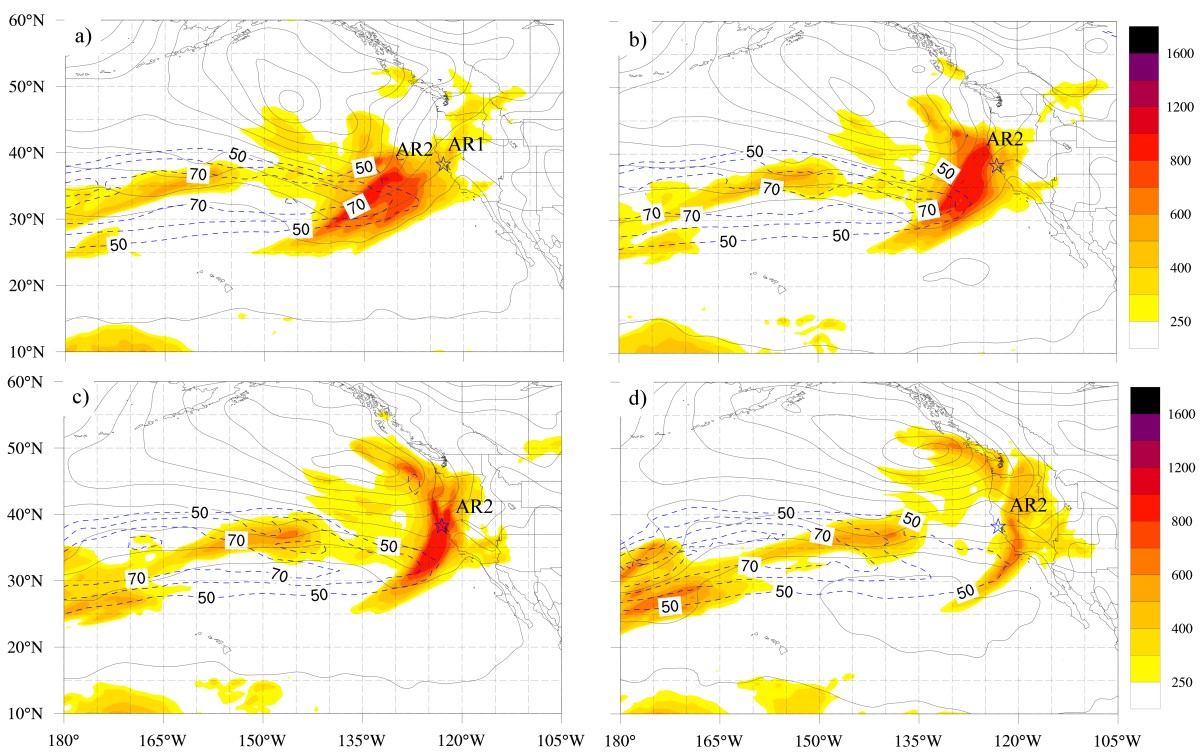

**Figure 3.** a) CFSR derived IVT ($\mathrm{kg\,m^{-1}\,s^{-1}}$; colorfill), SLP (hPa; grey contours every 5 hPa from 960 hPa) and jet layer horizontal wind isotachs ($\mathrm{m\,s^{-1}}$; blue dashed contours) valid at 12 UTC, 5 March 2016. b) as in a; except valid at 18 UTC, 5 March 2016. c) as in a; except valid at 00 UTC, 6 March 2016. d) as in a, except valid at 06 UTC 6 March 2016.





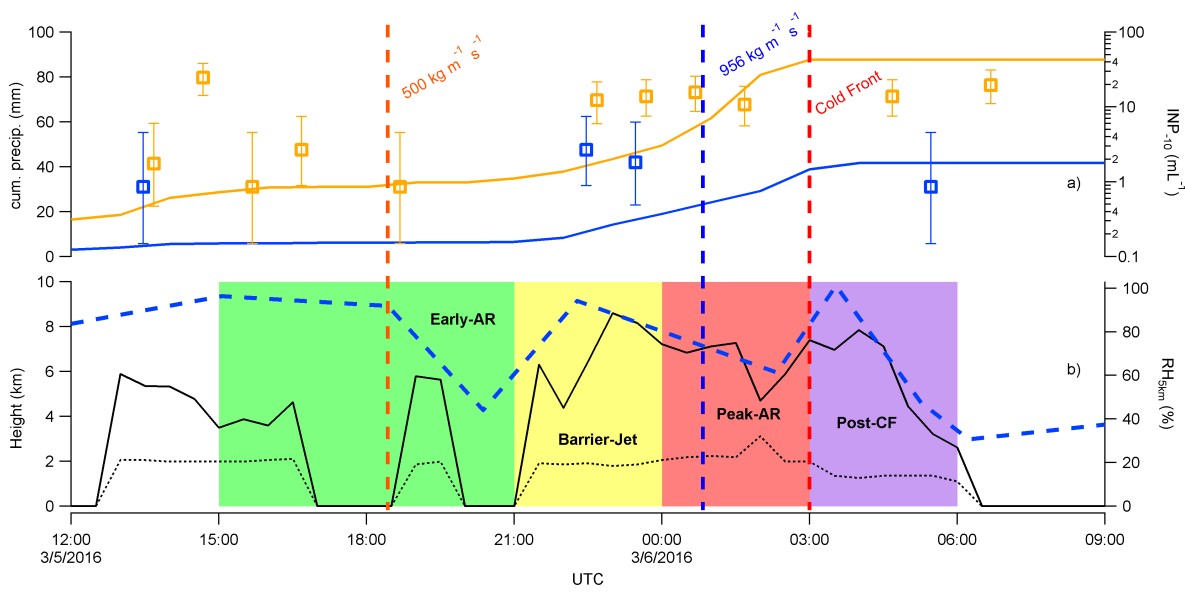

**Figure 4.** a) Timeseries of $\mathrm{INP}_{-10}$ $(\mathrm{mL}^{-1})$ at BBY (box-and-whisker - blue), at CZC (box-and-whisker - orange), accum. precip. (mm) at BBY (blue line) and accum. precip. at CZC (orange line). Timing of IVT surpassing $500\,\mathrm{kg\,m^{-1}\,s^{-1}}$ and cold front transit are annotated in red dashed lines. b) S-band radar derived echo-top (ET - black solid) and brightband (BB - black dashed) height (kmMSL) at CZC. Also shown is $RH_{5km}$ (%) from soundings (blue dashed). Shading depicts "Early AR", "Barrier Jet", "Peak AR", and "Post-CF" periods (section 4d), respectively.



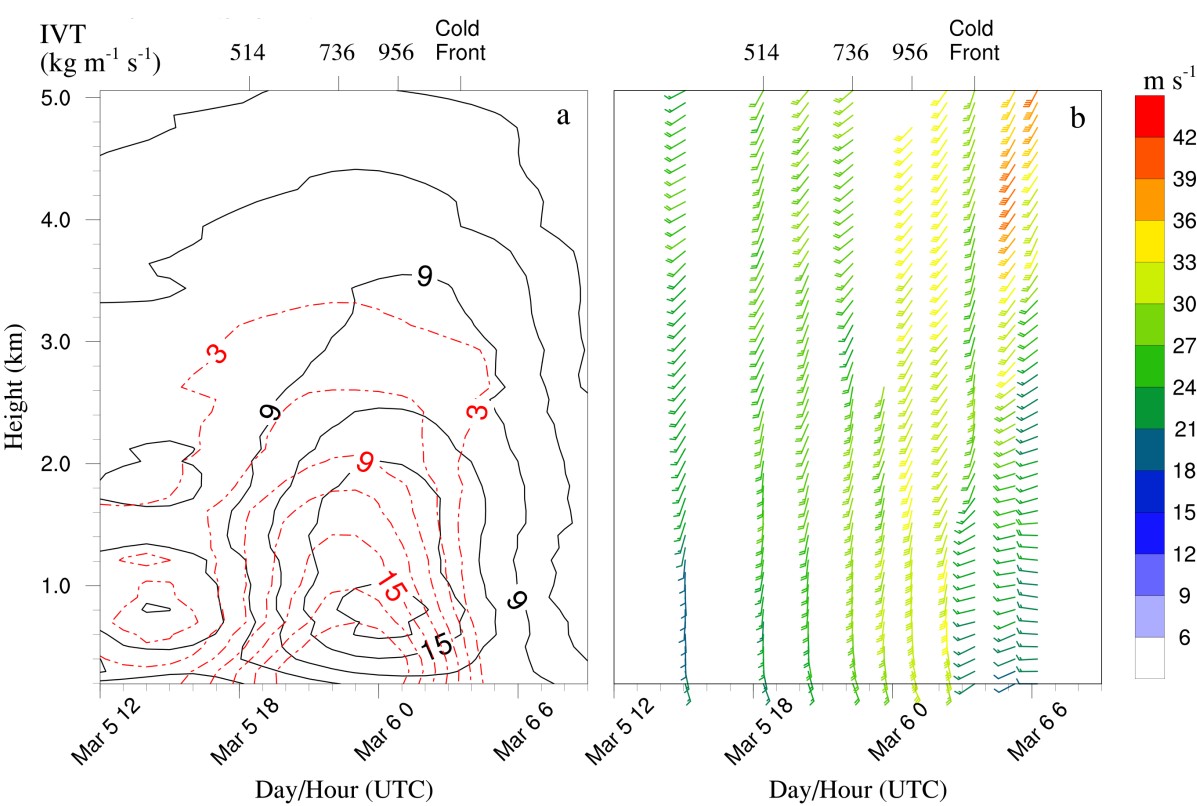

**Figure 5.** a) Upslope (black solid) and along-barrier (red dashed) water vapor flux $(\mathrm{g\,kg^{-1}\,s^{-1}})$ derived from rawinsondes during storm period. b) Rawinsonde horizontal wind profiles $(\mathrm{m\,s^{-1}}$, wind barbs colored by speed) during event. In each a and b, the time of significant sondes are marked along the top axis by their IVT $(\mathrm{kg\,m^{-1}\,s^{-1}})$ or by the arrival of the cold front.



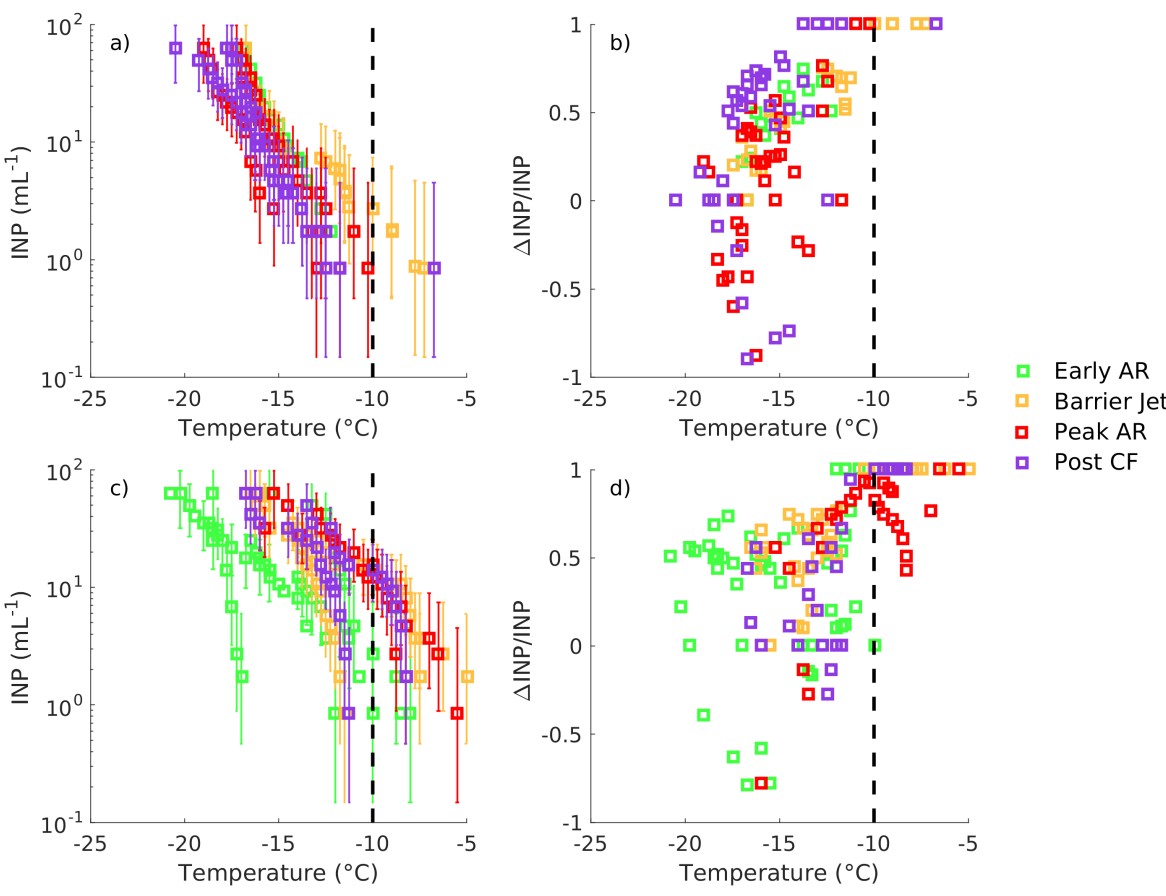

**Figure 6.** a) Un-heated $INP(T)$ (mL$^{-1}$) from BBY precipitation during "Early AR" (green), "Barrier Jet" (yellow), "Peak AR" (red), and "Post CF" (purple) periods. Whiskers denote technique standard error (mL$^{-1}$). b) as in a, except for $\Delta INP(T)/INP(T)$. c) as in a, except for un-heated precipitation samples from CZC. d) as in c, except for $\Delta INP(T)/INP(T)$.




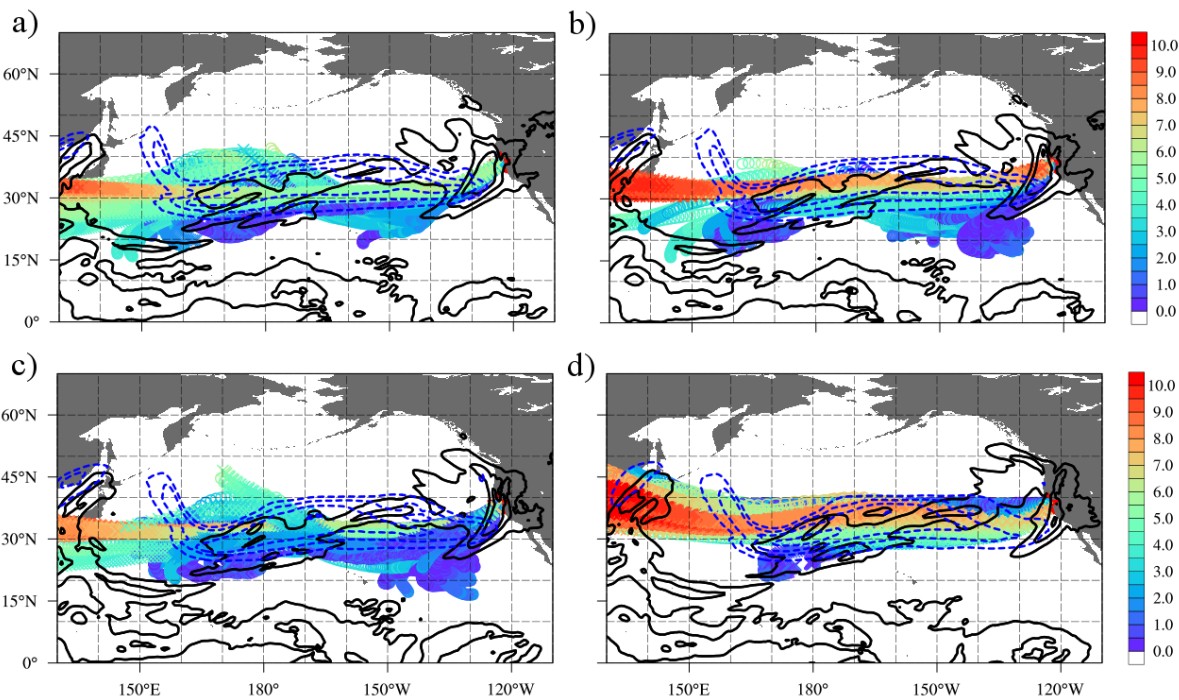

**Figure 7.** a) FLEXPART backward-simulated element position for releases from cloud-top ('X' markers) and mixed-phase ('O' markers) layers over CZC during Early AR period. Marker color denotes element altitude ($\mathrm{km\,AGL}$). Period average $IVT$ from CFS is shown by black contours from $250\,\mathrm{kg\,m^{-1}\,s^{-1}}$ to $750\,\mathrm{kg\,m^{-1}\,s^{-1}}$ every $250\,\mathrm{kg\,m^{-1}s^{-1}}$. Period average horizontal wind speed in the jet layer (see section 3a for layer definition) is shown by blue dashed contours from $50\,\mathrm{m\,s^{-1}}$ to $70\,\mathrm{m\,s^{-1}}$ every $10\,\mathrm{m\,s^{-1}}$. b) as in a, except for Barrier Jet period. c) as in a, except for Peak AR period. d) as in a, except for Post CF period.



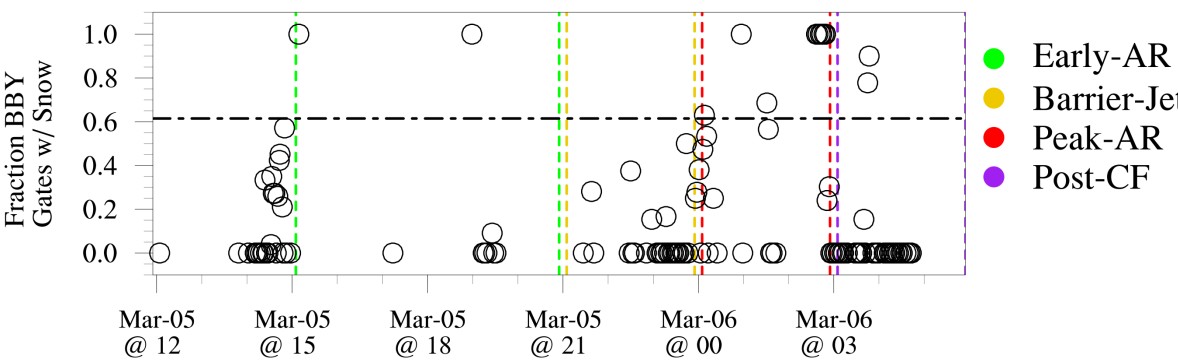

**Figure 8.** a) Timeseries of $P_{snow}^{BBY}$ (black circles) in the unblocked layer from all KDAX scans detecting precipitation at the BBY azimuth. The all-storm mean of $P_{snow}^{CZC}$ is shown by the horizontal dot-dash black line. Vertical dashed lines show the boundaries of the major storm periods, as coded by color in the legend.



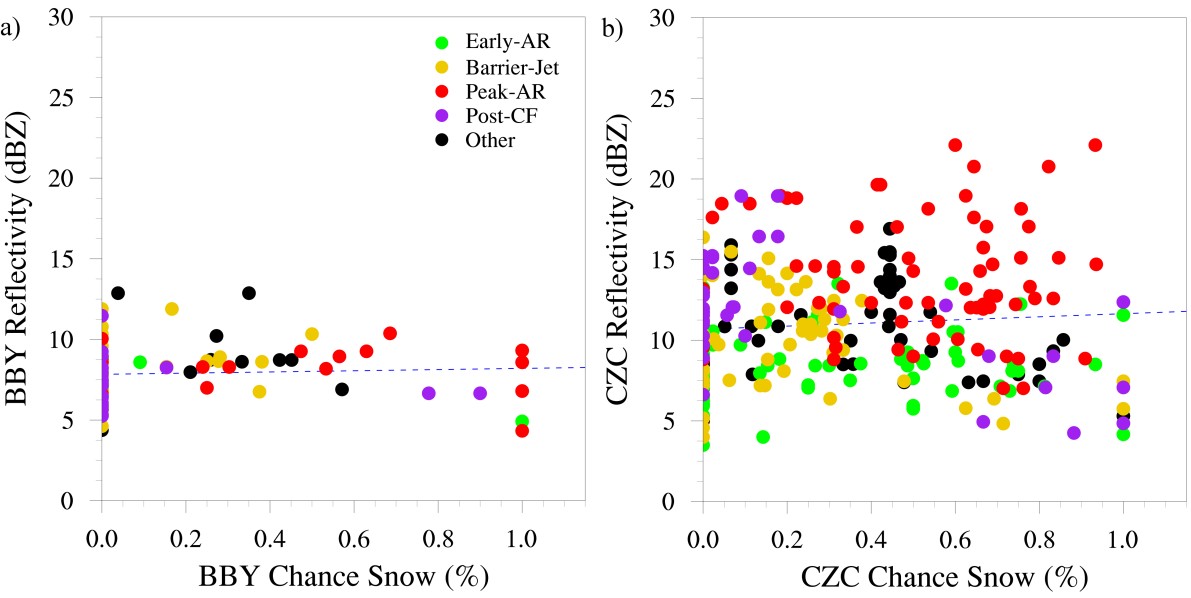

**Figure 9.** a) Relationship between $P_{snow}^{BBY}$ (ordinate) and BBY mean reflectivity (dBZ - abscissa) from all KDAX scans detecting precipitation at the BBY azimuth. Marker color depicts the major storm period each scan belonged to, as coded by color in the legend. b) as in a, except for $P_{snow}^{CZC}$ and CZC mean reflectivity (dBZ).





**Table 1.** ARO Measurements by site (BBY/CZC)

| Measurement | BBY | CZC | Reference |
|---|---|---|---|
| **449 MHz wind profiling radar** | X | | White et al. (2013) |
| **S-Band profiling precipitation radar** | | X | " " |
| **GPS-derived Integrated Water Vapor** | X | X | " " |
| **Surface weather station (rainguage, anemometer)** | X | X | " " |
| **ISCO 6712 water samplers** | X | X | http://www.teledyneisco.com/en-us/ |



**Table 2.** Balloon-borne soundings launched from BBY and their metadata: IVT, height of freezing isotherm, top (bottom) temperatures of the KDAX radar retrieval layer (see section 3c). Superscripts [M,C] denote maximum AR strength, transit of cold front, respectively

| Sounding time | $IVT$ (kg m$^{-1}$ s$^{-1}$) | $Z_{T\,=\,0°C}$ (m) | $T^{top}_{KDAX}$ (°C) | $T^{bot}_{KDAX}$ (°C) |
|---|---|---|---|---|
| 1504 UTC, 5 March 2016 | 416 | 2562 | -4.9 | -0.9 |
| 1826 UTC, 5 March 2016 | 514 | 2613 | -5.4 | -1.6 |
| 2022 UTC, 5 March 2016 | 560 | 2666 | -4.2 | -1.2 |
| 2217 UTC, 5 March 2016 | 736 | 2560 | -4.4 | -2.1 |
| 0050 UTC, 6 March 2016[M] | 956 | 2944 | -4.4 | 0.5 |
| 0220 UTC, 6 March 2016 | 922 | 2967 | -4.5 | 0.8 |
| 0332 UTC, 6 March 2016[C] | 553 | 2686 | -4.9 | -1.0 |
| 0516 UTC, 6 March 2016 | 467 | 2213 | -7.5 | -3.7 |
| 0614 UTC, 6 March 2016 | 314 | 2101 | -9.2 | -5.4 |





**Table 3.** Kinematic periods of AR2, their beginning and end time, maximum sounding-derived $IVT$, height of cloud layers (see section 3b) used for FLEXPART analysis, mean $INP_{-10}$ and accumulated precipitation at each site.

| Period name | Start time (UTC) | Max IVT (kg m$^{-1}$ s$^{-1}$) | $Z_{T=0^\circ C}$ / $Z_{T=-12^\circ C}$ / ETH (m MSL) | CZC Mean $INP_{-10}$ / CI$^-$ – CI$^+$ (mL$^{-1}$) | Accum. precip. (mm) (BBY / CZC) |
|---|---|---|---|---|---|
| Early AR | 15 UTC, 5 Mar | 560 | 2550 / 4800 / 5800 | 0.87 / 0.23 - 3.29 | 4.5 / 11.2 |
| Barrier Jet | 21 UTC, 5 Mar | 736 | 2550 / 4850 / 8600 | 8.71 / 4.5 - 14.9 | 7.6 / 10.4 |
| Peak AR | 00 UTC, 6 Mar | 956 | 2950 / 4850 / 7800 | 8.79 / 4.75 - 14.82 | 15.0 / 37.6 |
| Post CF | 03 UTC, 6 Mar | 553 | 2100 / 4150 / 8300 | 4.62 / 2.52 - 7.72 | 6.6 / 12.5 |





**Table 4.** Probability of instantaneous element residence in features of interest $P_{res}$, during FLEXPART backward simulation given a element arrived in the labelled period and layer. Non-zero $P_{res}$ are **bold**.

| Feature: | Period and Layer (mixed-phase: MP; cloud-top: CT) | | | | | | | |
|---|---|---|---|---|---|---|---|---|
| | Early AR | | Barrier Jet | | Peak AR | | Post CF | |
| | MP | CT | MP | CT | MP | CT | MP | CT |
| $P_{UTJ}$ | 0.0 | **0.003** | 0.0 | **0.194** | 0.0 | **0.028** | **0.04** | **0.235** |
| $P_{AR}$ | **0.351** | **0.231** | **0.411** | **0.033** | **0.452** | **0.194** | **0.290** | **0.075** |
| $P_{TBL}$ | 0.0 | 0.0 | **0.062** | 0.0 | **0.083** | 0.0 | **0.044** | 0.0 |
| $P_{MBL}$ | **0.158** | **0.172** | **0.300** | 0.0 | **0.398** | **0.182** | **0.313** | **0.028** |