# Peer review of "Contrasting Local and Long-Range Transported Warm Ice-Nucleating Particles During an Atmospheric River in Coastal California, USA"

_Atmospheric Chemistry and Physics, 2018_

## Short Comment (SC1) · 13 Aug 2018

It is well known that decomposed vegetation is a prolific source of atmospheric ice nuclei active at warm temperatures. There is a strong possibility that the active ice nuclei you observed and attributed to long-range transport from terrestrial sources are such nuclei. Two earlier studies discussing these sources are:

Schnell, R. C. and G. Vali: World-wide Source of Leaf-derived Freezing Nuclei, Nature, 246, 212,1973.

Schnell, R. C. and Vali, G.: Biogenic ice nuclei: part I. Terrestrial and marine sources,

J. Atmos. Sci. 33, 1554–1564, 1976.

**[ACPD](ACPD)**

---

## Referee Comment (RC1) · Anonymous Referee #1 · 29 Aug 2018

General Comments:

The manuscript discussed warm ice-nucleating particles during an atmospheric river in a California coast area. The study exemplified a comprehensive investigation of an important research topic: sources of ice nucleating particles that could initiate freezing at -10°C or warmer temperatures. This research deployed parallel sampling at two nearby sites that are only separated by 35 km but are destined to experience different ice nuclei sources at certain conditions. The study employed an automated ice spectrometer and multiple meteorological instruments carried by balloon soundings, as well as remote sensing techniques. Therefore, it contains valuable data and analysis that deserve to be published for the public's interests. The study also carried out complementary modelling and statistical analysis while observation data is unavailable, constituting a complete effort of comprehensive investigation. The results of the study presented some good proposals on future study of marine INPs. The manuscript is presented in a way that the authors pose logically relevant questions for the readers and then cope with these questions one after one till satisfaction. Although the organization of the manuscript is different from a normal research article, the logic can be well understood and should be acceptable. However, there are still some issues to be fixed for the manuscript to become publishable.

Specific Comments:

1, Marine-sourced particles are determined to be insignificant as warm INPs based on precipitation samples from these two sites during this AR. Aerosol particle concentration over ocean at cloud level is usually much lower than over land, but that doesn't prevent marine clouds contain more warm ice nuclei than terrestrial clouds because IN/CN ratio can be as low as 1 in 10e6 if marine aerosol are more efficient in ice nucleation. Since previous researches have reported the capability of marine sourced particles to serve as warm INPs, the authors should then eliminate the possibility of experimental artifacts (including sampling and AIS testing) that might potentially exclude marine particles. It would be very interesting to know if there is signature of marine-sourced particles in the precipitation samples from one or both sites. For the conclusions of this article to be solid, the most important artifact or mechanism to preclude is that marine particles didn't survive AIS analysis for samples from both sites. Does rainfall intensity have any impacts on INP concentration measured from precipitation samples?

2, Some statistical analysis in the article doesn't seem to be convincing. For example, in the last paragraph of section4.8, a linear regression is still discussed when R square is lower than 0.01. In this case, maybe there is no relationship to seek.

3, The summary section seems to be slightly lengthy and could be presented in a more

concise manner. It is understandable that the authors intend to present a summarized study in a logically sound manner. However, some of the detailed analysis could be taken from the summary section without influencing the integrity of the manuscript.

4, There are some typos and imperfection of abbreviations or acronyms, as well as room for grammar improvement. All these will be presented in technical corrections.

Technical corrections:

1, In abstract, ice nucleating particles are abbreviated as "INP", while it is abbreviated as "INPs" in the introduction section and other parts of the article. To be consistent within the article and with other published research articles in this topic, it is preferred to use "INPs". To use it as a general term, the authors should still be able to use "ice nucleating particle" or "INP" without further defining it.

2, There are a series of occurrences that the paper mentioned section 3a, 3b,3c, 3d, 4d, 4e, 4f etc., which the paper is constructed as sections 3.1, 3.2, 3.3, etc. These mismatches should be fixed completely using "find and replace" function of the document editing software.

3, There are some word choice discrepancy throughout the manuscript. For example, "timeseries" and "time series" are used interchangeably. It would be preferred to be consistent in one manuscript. Most of the articles used "MSL" for altitude, while there are two occasions of using "AGL", i.e., line 15 in section 4.1, and title 7. Authors should examine if these two usages are correct and if they can be expressed as "MSL" accordingly.

4, Comments about quality of Figure 1: In figure title, along-slope direction wind should be symbolized as "â" instead of "û", which is for upslope wind. In Fig. 1a, the circle and square marks are too large compared with the marks in Fig. 2a. It would be less confusing if the unit "m" was assigned to colorbar title instead of the subfigure title, since there is already a unit (arc-sec.). In Fig. 1b, the colorbar seems to contain one

single color in one small square, unlike the gradual change color scale in Fig. 1a. Therefore, the number should be aligned with the middle of the small square instead of the edge. Readers have to match "17" to "ocean" to determine that the numbers in the colorbar correspond to the colors above the numbers. It seems that the white/clear color in the bottom of colorbar is not used in the map and could be deleted. In that case, it would be unequivocal even if numbers sit at the bottom of each square.

5, In Figure 3, CFSR is not used or defined in the article elsewhere, it should be defined. Also, authors should be clear about what the star marks in the plot mean. In the figure 3 title, there are two usage of semicolon after "as in a", which could be replaced by comma.

6, In Figure 4, "acuum. precip." is used in the figure title while "cum. precip." is used in the ordinate (Y-axis) title of 4a. It is preferable to be consistent.

7, In Figure 7, the "X" and "O" marks are mentioned in the title but invisible in the figure. As mentioned in the second comment of technical corrections, it also contains section 3a but no such section can be found in the article. It should be changed to section 3.1 if that is what it means.

8, In Figure 9, the title confused ordinate with abscissa. Precipitation chance is abscissa (X axis) while reflectivity is ordinate (Y axis). Abscissa was also mistaken as ordinate in Figure 2 title. Actually only BBY seems to land on X-axis, so it is better to mention neither ordinate nor abscissa for Figure 2.

9, Throughout the manuscript, figures are referred inconsistently. Majority of the cases are referred to as Fig. X (where X is a number or number plus letter), while some are figure X, (e.g. "Figure 7" in Page 12 line 31 and "figure 7" in Page 12 line 34; Fig 4a in Page 16 Line 17). There examples are not exhaustive.

The following corrections are suggestions on grammar improvement and vocabulary selection.

Page 1 line 7: "Precipitation was collected at two sites, one coastal and one inland, that are separated by less than 35 km" –> "Precipitation samples were collected at two sites, one coastal and one inland, which are separated by about 35 km".

Page 1 line 13: "...warm INP are observed..." –> "...warm INPs were observed...". It seems the abstract uses past tense for other similar expressions.

Page 2 line 4: "including in the US state of California" –>"including the US state of California"

Page 2 line 19: "bacterium" –>"bacteria", unless there is a reason to explain why "virus" was used as plural but bacterium was used as singular.

Page 4 line 11: "raingauge" –> "rain gauge", refer to line 22 on the same page

Page 4 line 28: " at irregular interval"–>"at irregular intervals"

Page 5 line 24: "..to be bio-INP" –>"...as bio-INP"

Page 5 line 29-30: "A short definition of each and identification methodology using study datasets is to follow" –> A short definition for each of them and/or their identification methodology are to follow"

Page 6 line 19: "along-slope"–>"Along-slope"

Page 6 line 25-30: MBL and TBL seem to be defined awkwardly. Can they be better defined even though the current definitions are unambiguous?

Page 10 line 9: "The remainder of this study will focus on AR2" –> "The remainder of this paper will focus on AR2".

Page 31-34, Table 1 and Table 2 don't have a dot at the end of their titles while Table 3 and Table 4 have.

---

## Referee Comment (RC2) · Anonymous Referee #2 · 24 Sep 2018

**General Comments:**

The paper describes the measurement of ice nucleating particles (INPs) in precipitation collected during an atmospheric river event at two ground measurement stations in California – a coastal station (BBY) and an inland site (CZC). The central conclusion of the paper is that warm INPs (i.e., those that initiate freezing at termperatures warmer than -10 deg. C) are found to increase during the atmospheric river event at CZC, but no change is detected at BBY. FLEXPART trajectory modeling and radar measurements of cloud properties are used to provide context for the atmospheric state during the INP measurements, although the relevance of this ancillary modeling

and measurements to the central conclusion of the paper is not clear. Overall, the manuscript is quite lengthly with extensive discussion of the meteorological evolution of the atmospheric river. However, it is not clear to me how these details support the proposed science questions and the strong conclusions that are reached. Rather, it seems that the primarily piece of evidence to support the role of terrestrial aerosol as "an important source of warm INP during this atmospheric river" is that the CZC is inland and the BBY site is coastal, so any difference between the two must be caused by the intervening land surface. FLEXPART modeling shows that probability of trajectory air parcels residing within the terrestrial boundary layer is zero during the early part of the atmospheric river and "small, but non-zero" during the latter portions of the atmospheric river. Is such a small residence time sufficient to explain the marked, ten-fold increase in warm INPs? Also, the transition in language from the measured "small, but non-zero" conclusion on Pg. 16, Line 29-31 to "important sources of warm INP" on Pg. 1, Line 16-17 seems disingenuous. In sum, the authors ask important and bold science questions (Pg. 3, Lines 26-31), but the limited amount of data from this one case study and the weak interpretation of the FLEXPART and radar data do not support the authors' proposed answers to these questions. The manuscript length, lack of adequate description of some introduced quantities, and confusing internal referencing significantly detract from the readability of the manuscript. The data and results may be of interest to the readers of Atmos. Chem. Phys. as a much shorter discussion of an interesting case study; however, I do not think the results presented here allow the authors to convincingly address the proposed science questions. Therefore, I recommend that the manuscript be extensively revised, shortened, and reframed as a case study analysis before I could recommend it as suitable for publication.

**Specific comments:**

1) I don't understand the discussion on Pg. 3, Lines 22-25. How does the blocking of the radar return by the coastal mountain range ensure that hydrometeor information is indicative of mixed phase clouds? I get that this limits the radar signal to roughly 2.9-3.6 km in altitude. Are the authors saying that freezing conditions do not exist below 2.9 km? Similarly, is the temperature at 3.6 km always above -10 deg. C? From Table 2, it appears that there is a great deal of variability regarding the extent of the radar retrieval layer top and bottom.

2) There is insufficient detail provided in Section 3.1. What are the signficance of BBH and ETH? A brief description of how these quantities are obtained should be included so that the reader doesn't have to search out the reference citation to understand what they are. How and why are these data being using in this study? Also, why do we care about the LLJ, CBJ, and polar cold front? Here, the CBJ is described as a feature, while in Figure 4, the barrier jet is denoted as a time period. Basically, Pg. 6 is a laundry list of different parameters, but some additional context of why these parameters are important and how they are / will be used would be very helpful here.

3) The references to sections are confusing as all sections are numeric, while some references use letters. Presumably, 3a = 3.1, 3b = 3.2, etc. Regardless of that minor technical fix, the pointers included in a lot of places are very vague. For example, what are "significant kinematic features" in Section 3.1? Does it make sense to say that 2000 elements were released per layer for three consecutive hours surrounding the coastal barrier jet? What is meant by a kinetmatic feature (Pg. 7, Line 8)? On Pg. 7, Line 29, it's stated that the methods in Sections 3.1-3.3 are used to link INP source regions to clouds over BBY and CZC via means of FLEXPART simulations, but Section 3.1 is largely definitions. All of this internal referencing is very confusing and detracts from, rather than helps, readability.

4) What is the meaning of the sentences on Pg. 8, Lines 1-3: "...we can identify proxy regions for local INP sources using the terrestrial and marine boundary layers, but these methods cannot capture all possible LRT source regions. Thus, we must in part

make inferences about source after rejecting alternate hypotheses if the mechanisms examined are not supportive." What are these alternate hypotheseses and mechanisms?

5) The paragraph on Pg. 8, Lines 22-29 is very confusing and needs to be revised to be clearer. What is meant by the statement that the authors "sought to preserve the mixed phase temperature range as found by the soundings in Table 2"? Why are Chi-Square independence tests being performed? Why is a rule of thumb bing applied to the minimum expected population? The application of these statistical methods here (and throughout the manuscript) are not well described, and I don't understand why they should be done and are being done.

6) The discussion on Pg. 10, Lines 11-18 doesn't seem to match the graph. INP-10 at CZC seems to be between 1-4/mL on March 5th (where does 0.25/mL come from?). Similarly, on March 6, INP-10 at CZC are 10-15/mL (where does 3/mL come from?). Since there's only a few data points for BBY, I don't think it can be stated that "BBY only occasionally neared 2/mL".

7) Why are there so few data points for INP-10 in Figure 4? Do all of the time periods where there are no data points reflect that the concentration of INP-10's is below the detection limit? What is the detection limit? Points that are zero or below the lower limit of detection need to be added to the graph as well in order to evaluate trends. Otherwise, statistical and interpretative significance might erronously be applied to only a handful of otherwise insignificant data points.

8) On Pg. 10, Lines 23-25, it's stated that there are not precipitating hydrometeors during 15-21 UTC on March 5th, but it looks like the cummulative precipitation curves increase during this period. How can it be both ways?

9) The discussion on Pg. 12, Lines 5-9 is all highly speculative and not supported by any evidence in this manuscript. Please revise or strike this paragraph/conclusion.

10) What are the more exotic functions of temperature used/referenced on Pg. 12, Line 13? I don't understand how the authors are able to state that "it is likely that biological material contributed significantly to INP concentrations for T < -10 deg. C at CZC, but not at BBY." Where is the evidence!?

11) Where in Section 3.1 is it stated that the jet stream is located between altitudes of 6.5 and 11 km MSL as implied on Pg. 12, Line 33?

12) The sentence on Pg. 13, Line 14, "Table 4 presents the probability of element residence (section 3c) in the UTJ, AR, MBL, and TBL." is another example of sloppy internal referencing. Why is Section 3.3 being invoked here?

13) In the conclusions on Pg. 16, it is stated that terrestrial warm INPs are abundant and that marine warm INPs are not evident, but there are warm INP data points reported for BBY in Figure 4. If these are not marine warm INPs, where do they come from?

14) If the small, non-zero change in instantaneous element residence in the terrestrial boundary layer is really the driver of why the warm INP concentrations vary at CZC, then why do the INP concentrations not vary with the varying numbers shown in Table 4 – 6.2

**Technical corrections:**

Pg. 6, Line 4: IVT is not yet defined. It is defined on Pg. 9, Line 10, but only in passing.

Pg. 9, Line 15: Reference to Martin et al. seems out of place

Pg. 9, Line 18: Figure 5b is reference out of order

Pg. 12, Line 17: The reference to Section 2d (2.4) does not seem right. Should this be 2.5?

---

## Author Comment (AC1) · 15 Dec 2018

Contrasting Local and Long-Range Transported Warm Ice-Nucleating Particles During an Atmospheric River in Coastal California

Authors' Response (AR) to Public and Referee Comments

Note: All references in the Authors responses can be found in the references section of the manuscript acp-2018-702 or in the supplemental material, as applicable.

Public Comments SC1: It is well known that decomposed vegetation is a prolific source of atmospheric ice nuclei active at warm temperatures. There is a strong possibility

that the active ice nuclei you observed and attributed to long-range transport from terrestrial sources are such nuclei. Two earlier studies discussing these sources are: Schnell, R. C. and G. Vali: World-wide Source of Leaf-derived Freezing Nuclei, Nature, 246, 212,1973. Schnell, R. C. and Vali, G.: Biogenic ice nuclei: part I. Terrestrial and marine sources, J. Atmos. Sci. 33, 1554-1564, 1976.

AR-SC1: The authors agree that it is possible for long-range transported bioparticles originating as decomposed vegetation to influence precipitation formation at our study location. The above references and a brief discussion of this possibility has been added on page 2, lines 17-20.

Comments from Referee One: RC1: "Marine-sourced particles are determined to be insignificant as warm INPs based on precipitation samples from these two sites during this AR. Aerosol particle concentration over ocean at cloud level is usually much lower than over land, but that doesn't prevent marine clouds contain more warm ice nuclei than terrestrial clouds because IN/CN ratio can be as low as 1 in 10e6 if marine aerosol are more efficient in ice nucleation. Since previous researches have reported the capability of marine sourced particles to serve as warm INPs, the authors should then eliminate the possibility of experimental artifacts (including sampling and AIS testing) that might potentially exclude marine particles. It would be very interesting to know if there is signature of marine-sourced particles in the precipitation samples from one or both sites. For the conclusions of this article to be solid, the most important artifact or mechanism to preclude is that marine particles didn't survive AIS analysis for samples from both sites.

AR-RC1: The authors acknowledge that droplet freezing in the AIS is agnostic to the source of the INP, thus during times when freezing events are recorded for T > -10 C, the AIS alone cannot determine whether the freezing nuclei originated as marine or terrestrial particles. Further, the authors did not possess a laboratory method to analyze the source of individual freezing nuclei or to analyze the source of the collected population of freezing nuclei in a manner separate from the ambient particle population. The latter methodological limitation is especially problematic given the salient point raised by the referee: IN/CN ratio can be as low as 1 in 10e6. Accordingly, even trends in the ambient particle population source will shed no additional light on the source of the freezing nuclei. We thus principally relied on the contrast between the coastal and the inland sites and the additional meteorological analyses to make inferences in the source of warm INP. However, we do agree with the referee's assessment that it is important to exclude the possibility that the collection or AIS methods preferentially destroyed marine INP at the coastal and not at the inland site. To address the former, we have written additional details regarding care taken during sample collection and the operation of the AIS. See page 5 , lines 9-10. To address the latter, we have provided supplemental material (SM) covering analysis of precipitation sample insoluble residues performed using the aerosol time-of-flight mass spectrometer (ATOFMS). SM includes a methodological description including sample preparation, nebulizing, injection, and basic concepts of ATOFMS operation and particle identification. ATOFMS was used to classify single insoluble residue particles into four separate types, including a bioparticle type. The particle classification method and bioparticle type have been published in previous studies, with references provided in the SM. Figure SM1 shows that the bioparticle type was the most numerous at both coastal and inland sites during all kinematic periods. While we cannot separate these bioparticles according to their marine or terrestrial sources, their ubiquity and similar concentration at both sites suggest that a significant number are from marine sources. These bioparticles are related to warm INP in that all freezing events triggered for T > -10 C in the AIS should be caused by insoluble residue bioparticles, but not all insoluble residue bioparticles are capable of triggering freezing in the AIS (e.g. IN inactive bioparticles). Thus, Figure SM1 demonstrates that marine bioparticles were collected and preserved for laboratory analysis from both sites, while the low number warm INP collected at the coastal site was a result of the inability of the marine bioparticles collected there to trigger freezing events at T > -10 C.

The above discussion has been partially reproduced in-text, see page 10, lines 3 – 16.

RC2: Does rainfall intensity have any impacts on INP concentration measured from precipitation samples?

AR-RC2: Yes, as found by authors such as Huffman et al., 2013; Prenni et al., 2013; Morris et al., 2014; Bigg et al., 2015 (see references), rainfall intensity can impact INP concentration in precipitation by stimulating local emission of bio-INP, the so-called "bioprecipitation feedback". See page 2, lines 21 – 24 for discussion. We note that bioprecipitation feedback would not cause the type of artifact that the referee has asked about and we are not aware of other rainfall intensity impact that would.

RC3: Some statistical analysis in the article doesn't seem to be convincing. For example, in the last paragraph of section4.8, a linear regression is still discussed when R square is lower than 0.01. In this case, maybe there is no relationship to seek.

AR-RC3: The authors' intent in discussing the least-square relationship in the final paragraph of section 4.8 was to further emphasize that any relationship between precipitation rate and likelihood of detecting snow hydrometeors was unlikely. We agree that quoting the R-Square value is sufficient and that the additional least-square discussion is unnecessary. It has been redacted, see page 15, lines 4 – 5.

RC4: The summary section seems to be slightly lengthy and could be presented in a more concise manner. It is understandable that the authors intend to present a summarized study in a logically sound manner. However, some of the detailed analysis could be taken from the summary section without influencing the integrity of the manuscript.

AR-RC4: The authors agree with this assessment and have reduced the amount of detailed analysis presented in the discussion. C.f. page 15, lines 24 – 34.

RC5: There are some typos and imperfection of abbreviations or acronyms, as well as room for grammar improvement. All these will be presented in technical corrections.

AR-RC5: The authors thank the referee for diligence in improving the grammatical and technical writing content of this article. Changes have been made as requested, and

will be itemized below in the responses to technical corrections.

Referee One Technical Corrections: RC6: In abstract, ice nucleating particles are abbreviated as "INP", while it is abbreviated as "INPs" in the introduction section and other parts of the article. To be consistent within the article and with other published research articles in this topic, it is preferred to use "INPs". To use it as a general term, the authors should still be able to use "ice nucleating particle" or "INP" without further defining it.

AR-RC6: This has been changed to INPs in the abstract, see page 1, line 1.

RC7: There are a series of occurrences that the paper mentioned section 3a, 3b,3c, 3d, 4d, 4e, 4f etc., which the paper is constructed as sections 3.1, 3.2, 3.3, etc. These mismatches should be fixed completely using "find and replace" function of the document editing software.

AR-RC7: Internal references to subsections have been fixed to follow the ACP outline style 3.1, 3.2, 3.3, etc. See for example page 4, line 12.

RC8: There are some word choice discrepancy throughout the manuscript. For example, "timeseries" and "time series" are used interchangeably. It would be preferred to be consistent in one manuscript. Most of the articles used "MSL" for altitude, while there are two occasions of using "AGL", i.e., line 15 in section 4.1, and title 7. Authors should examine if these two usages are correct and if they can be expressed as "MSL" accordingly.

AR-RC8: Instances of timeseries have been updated to comply with American Meteorology Society Glossary spelling time series. The former use of AGL was intentional and correct. The latter use, in the caption of Figure 6, has been changed to MSL.

RC9: In Figure 9, the title confused ordinate with abscissa. Precipitation chance is abscissa (X axis) while reflectivity is ordinate (Y axis). Abscissa was also mistaken as ordinate in Figure 2 title. Actually only BBY seems to land on X-axis, so it is better to

mention neither ordinate nor abscissa for Figure 2.

AR-RC9: These incorrect references have been fixed. See caption of Figs. 2 and 9, respectively.

RC10: Throughout the manuscript, figures are referred inconsistently. Majority of the cases are referred to as Fig. X (where X is a number or number plus letter), while some are figure X, (e.g. "Figure 7" in Page 12 line 31 and "figure 7" in Page 12 line 34; Fig 4a in Page 16 Line 17). There examples are not exhaustive.

AR-RC10: All internal references to a figure have been updated such that the convention "Fig." is used, except at the beginning of a sentence, where "Figure" is used.

RC11: Precipitation was collected at two sites, one coastal and one inland, that are separated by less than 35 km

AR-RC11: Has been changed as suggested to "Precipitation samples were collected at two sites, one coastal and one inland, which are separated by about 35 km". See page 1, lines 7-8.

RC12: …warm INP are observed… It seems the abstract uses past tense for other similar expressions.

AR-RC12: Has been changed as suggested to "warm INPs were observed". See page 1, line 14.

RC13: …including in the US state of California.

AR-RC13: Has been changed as suggested to "including the US state of California." See page 2, line 4.

RC14: "…bacterium…". Singular is incorrect in this context.

AR-RC14: Has been changed as suggested to "bacteria". See page 2, line 19.

RC15: "…rainguage…". Inconsistent with rain gauge from page 4, line 20.

AR-RC15: All instances of rainguage, rain-gauge, or variants have been changed to match the American Meteorological Society Glossary spelling "rain gauge". See page 4, line 9 for example.

RC16: "…at irregular interval…" on page 4, line 26 has been changed to "…at irregular intervals…".

RC17: "…to be bio-INP" on page 5, line 22 has been changed to "…as bio-INP".

RC18: "A short definition of each and identification methodology using study datasets is to follow" has been redacted following AR-RC23.

RC19: "along-slope" on page 7, line 19 has been changed to "Along-slope".

RC20: MBL and TBL seem to be defined awkwardly. Can they be better defined even though the current definitions are unambiguous?

AR-RC20: The definitions have been updated to read Marine boundary layer (MBL): The MBL was defined by all locations where CFS geopotential height (m MSL) is less than the FLEXPART planetary boundary layer depth and the latitude and longitude are over the Northeast Pacific Ocean. Terrestrial boundary layer (TBL): The TBL was defined similarly to the MBL, expect latitude and longitude must have been over the US state of California. See page 7, lines 31 – 32 and page 8, lines 1-2.

RC21: "The remainder of this study will focus on AR2" on page 10, line 9..

AR-RC21: This passage has been redacted following AR-RC24.

RC22: Table 1 and Table 2 don't have a dot at the end of their titles while Table 3 and Table 4 have.

AR-RC22: Table 1 and Table 2 titles have been fixed accordingly.

Comments from Referee Two: General Comments RC23: FLEXPART trajectory modeling and radar measurements of cloud properties are used to provide context for the

atmospheric state during the INP measurements, although the relevance of this ancillary modeling and measurements to the central conclusion of the paper is not clear.

AR-RC23: Though the above comment comes from the summary and not the itemized lists provided by the referee, the authors take seriously the possibility that the relevance of FLEXPART and radar analysis is unclear. To help clarify, the authors have added additional motivation for these analyses in the methods section. Note that the additional motivation for FLEXPART and radar analysis helps to address later comments and the responses to those comments will reference AR-RC23. The relevant new methods section can also be found from page 5, line 23 through page 9, line 18.

RC24: . . . with extensive discussion of the meteorological evolution of the atmospheric river. However, it is not clear to me how these details support the proposed science questions and the strong conclusions that are reached.

AR-RC24: The authors acknowledge that more meteorological description of the event was provided than is necessary to understand the scientific goals. Meteorological background in former sections 4.1 and 4.2 have been shortened significantly. In addition, the synoptic meteorology figure (formerly Figure 3) has been redacted. Because the design of the FLEXPART modeling depends on the evolution of kinematic features above the sites and the definition of kinematic periods (e.g. Early AR, Barrier Jet, etc.), we retained this meteorological discussion (section 4.2).

RC25: FLEXPART modeling shows that probability of trajectory air parcels residing within the terrestrial boundary layer is zero during the early part of the atmospheric river and "small, but non-zero" during the latter portions of the atmospheric river. Is such a small residence time sufficient to explain the marked, ten-fold increase in warm INPs?

AR-RC25: The authors consider the small but non-zero residence in the continental boundary layer by cloud-inflowing air a necessary but insufficient condition to identify INP source region. Other evidence, such as the contrast between the coastal and

inland site, the heat treatment of precipitation samples prior to AIS measurement, and the analysis of radar retrievals over the coastal and inland sites serve as evidence to dismiss alternate hypothetical sources. To reassure the reader that relative contribution to cloud-inflowing air of 4 – 8 % can be consistent with a ten-fold increase in warm INP, we have provided the following contextual discussion on page 13, lines 32 – 34 and page 14, lines 1-9:

The reader may wonder whether it is reasonable for the warm INP content of precipitation to so strongly respond (order magnitude increase, see Fig. 4a) to the arrival of parcels from the terrestrial boundary layer during and after the Barrier-Jet period, given the fractional contribution of these parcels to the cloud-inflowing airmass is at most 8%. It is prudent to note that the ambient concentration of warm INP in the terrestrial boundary layer upstream of CZC is unknown, but work by (Huffman et al., 2013; Prenni et al., 2013; Morris et al., 2014; Bigg et al., 2015) demonstrate that ambient INP concentrations often rise dramatically in response to precipitation, thus we cannot use the FLEXPART analysis to estimate the increase in number concentration of cloud-inflowing warm INP of terrestrial origin. It has also been shown that approximately 1 in 106 condensation nuclei (CN) serve as IN in the troposphere, further underscoring the potential impact on clouds of relatively few INP. Finally, Stopelli et al. (2015) argue that INP are removed much more efficiently by precipitation than are other CN. We can thus expect that the precipitation INP content will respond in a highly non-linear fashion to changes in the ambient concentration of INP in cloud-inflowing air. Indeed, because the ice-phase microphysical processes governing removal of INP by precipitation may vary independently from airmass source, we need not expect the precipitation INP content to strongly covary with changes in terrestrial boundary layer residence.

RC-26: the transition in language from the measured "small, but non-zero" conclusion on Pg. 16, Line 29-31 to "important sources of warm INP" on Pg. 1, Line 16-17 seems disingenuous.

AR-RC26: The passage previously written on page 16, line 29-31 has been redacted

while addressing AR-RC4 and AR-RC27.

RC27: I recommend that the manuscript be extensively revised, shortened, and re-framed as a case study analysis before I could recommend it as suitable for publication.

AR-RC27: The authors have taken steps to revise and shorten the manuscript, see AR-RC4, AR-RC23 and AR-RC24 for examples. Previous length of the manuscript body was 16 pp plus 9 figures in ACP draft form, new length is 15 pp plus 8 figures. We have also added new language emphasizing that this paper describes a single case. See page 1, line 6. We have additionally narrowed the paper goals. The new goals now read 1. ÂăWhat roles do terrestrial, marine and LRT sources have in determining the warm INPs during this AR? 2. What are the transport and cloud injection mechanisms for each of these sources? 3. ÂăWhen warm INPs are present in precipitation, are cloud microphysics impacted?

Referee Two Specific Comments: RC28: I don't understand the discussion on Pg. 3, Lines 22-25. How does the blocking of the radar return by the coastal mountain range ensure that hydrometeor information is indicative of mixed phase clouds? I get that this limits the radar signal to roughly 2.9- 3.6 km in altitude. Are the authors saying that freezing conditions do not exist below 2.9 km? Similarly, is the temperature at 3.6 km always above -10 deg. C? From Table 2, it appears that there is a great deal of variability regarding the extent of the radar retrieval layer top and bottom.

AR-RC28: The authors are saying that for this storm event, freezing conditions do not extend significantly below 2.9 km (the bottom of the unblocked radar beam) and that the temperature at 3.6 km is always warmer than -10 deg. C. There is variability in the temperature measured at unblocked layer top and bottom (Table 2), but the intra-event variability is not critical to the method of radar analysis (see Methods sections 3.5 and 3.6). Rather it is critical that the temperature in the unblocked beam does not drop below -9.2 deg. C, and does not rise above 0.8 deg. C, as shown in Table 2. The information retrieved by the radar thus applies to hydrometeors in the temperature range

for warm INP activation. To clarify and be more precise, page 3, Lines 19 – 23 now read "We will additionally demonstrate that the temperature lapse rates of this storm and partial beam blocking by the coastal mountain range near the measurement sites constrained weather service radar such that retrieved signal from hydrometeors with temperatures -9.2 C < T <Åǎ0.8 C. The remotely sensed hydrometeors thus approximately overlap with the temperatures of warm INP activation."

RC29: There is insufficient detail provided in Section 3.1. What are the signficance of BBH and ETH? A brief description of how these quantities are obtained should be included so that the reader doesn't have to search out the reference citation to understand what they are. How and why are these data being using in this study? Also, why do we care about the LLJ, CBJ, and polar cold front? Here, the CBJ is described as a feature, while in Figure 4, the barrier jet is denoted as a time period. Basically, Pg. 6 is a laundry list of different parameters, but some additional context of why these parameters are important and how they are / will be used would be very helpful here.

AR-RC29: The authors see the potential for confusion in the manner that these parameters have been introduced. We addressed this in concert with AR-RC22 by reorganizing section 3 to begin with background and motivation for the FLEXPART modeling and radar analysis. Therein, we motivated the need to identify kinematic periods and significant layers. See page 6, lines 28 – 32.

RC30: The references to sections are confusing as all sections are numeric, while some references use letters. Presumably, 3a = 3.1, 3b = 3.2, etc. Regardless of that minor technical fix, the pointers included in a lot of places are very vague. For example, what are "significant kinematic features" in Section 3.1? Does it make sense to say that 2000 elements were released per layer for three consecutive hours surrounding the coastal barrier jet? What is meant by a kinetmatic feature (Pg. 7, Line 8)? On Pg. 7, Line 29, it's stated that the methods in Sections 3.1-3.3 are used to link INP source regions to clouds over BBY and CZC via means of FLEXPART simulations, but Section

3.1 is largely definitions. All of this internal referencing is very confusing and detracts from, rather than helps, readability.

AR-RC30: Please see the authors' response to reviewer 1, AR-RC7 for details regarding internal referencing of sections. Other sources of confusion, such as a definition for significant kinematic features and motivation for the FLEXPART methodology and its dependence on feature identification have been addressed in the reorganization of the methods section. See AR-RC23.

RC31: What is the meaning of the sentences on Pg. 8, Lines 1-3: "...we can identify proxy regions for local INP sources using the terrestrial and marine boundary layers, but these methods cannot capture all possible LRT source regions. Thus, we must in part make inferences about source after rejecting alternate hypotheses if the mechanisms examined are not supportive." What are these alternate hypotheseses and mechanisms?

AR-RC31: Hypothesis testing related to airmass source has been re-written. See new methods section and RC-22 for details.

RC32: The paragraph on Pg. 8, Lines 22-29 is very confusing and needs to be revised to be clearer. What is meant by the statement that the authors "sought to preserve the mixed phase temperature range as found by the soundings in Table 2"? Why are Chi-Square independence tests being performed? Why is a rule of thumb bing applied to the minimum expected population? The application of these statistical methods here (and throughout the manuscript) are not well described, and I don't understand why they should be done and are being done.

AR-RC32AR: The authors have revised this paragraph for clarity. See AR-RC23 and Page 8, lines 27 - 31 through page 9, lines 1 – 3.

RC33: The discussion on Pg. 10, Lines 11-18 doesn't seem to match the graph. INP-10 at CZC seems to be between 1-4/mL on March 5th (where does 0.25/mL come

from?). Similarly, on March 6, INP-10 at CZC are 10-15/mL (where does 3/mL come from?). Since there's only a few data points for BBY, I don't think it can be stated that "BBY only occasionally neared 2/mL".

AR-RC33: These points have been corrected accordingly, see page 10, lines 14 – 18.

RC34: Why are there so few data points for INP-10 in Figure 4? Do all of the time periods where there are no data points reflect that the concentration of INP-10's is below the detection limit? What is the detection limit? Points that are zero or below the lower limit of detection need to be added to the graph as well in order to evaluate trends. Otherwise, statistical and interpretative significance might erronously be applied to only a handful of otherwise insignificant data points.

AR-RC34: The detection limit for the AIS has been added to section 2.5 on page 5, lines 15 – 16. The detection limit for each sample has been annotated to Fig. 4 wherever the reading was below detection limit.

RC35: On Pg. 10, Lines 23-25, it's stated that there are not precipitating hydrometeors during 15-21 UTC on March 5th, but it looks like the cummulative precipitation curves increase during this period. How can it be both ways?

AR-RC35: The authors acknowledge this inconsistency in reporting. The passage on page 10, line 22 has been changed to read "S-Band retrievals are intermittently missing between 15 UTC and 21 UTC on 5 March."

RC36: The discussion on Pg. 12, Lines 5-9 is all highly speculative and not supported by any evidence in this manuscript. Please revise or strike this paragraph/conclusion.

AR-RC36: These lines have been removed as requested.

RC37: What are the more exotic functions of temperature used/referenced on Pg. 12, Line 13?

AR-RC37: The authors were referring to graphs produced in the referenced articles

(e.g. Petters and Wright, 2015), and thus did not intend to provide a specific functional form. To avoid confusion, we have changed the passage on page 11, line 19 to read "that cannot be modeled by a simple log-linear temperature relationship."

RC38: I don't understand how the authors are able to state that "it is likely that biological material contributed significantly to INP concentrations for T < -10 deg. C at CZC, but not at BBY." Where is the evidence!?

AR-RC38: The authors agree that this statement more is more defensible if it is written after the discussion of the fractional change in INP after sample heating discussion in the following paragraph. Also, the analysis therein shows that heat-sensitive material (inferred to be biological) did contribute to INP concentrations at BBY for a few samples. The passage has been moved and modified to read "Heat treatment and INP(T) functional form support the conclusion that biological material contributed to warm INP concentrations at CZC for most samples. However, biological material contributed to warm INP concentration at BBY only for a few samples." See page 11, lines 32 - 34.

RC39: Where in Section 3.1 is it stated that the jet stream is located between altitudes of 6.5 and 11 km MSL as implied on Pg. 12, Line 33?

AR-RC39: This section number has been changed as part of AR-RC23, The explanation for jet stream altitudes is now on page 8, lines 3 - 7.

RC40: The sentence on Pg. 13, Line 14, "Table 4 presents the probability of element residence (section 3c) in the UTJ, AR, MBL, and TBL." is another example of sloppy internal referencing. Why is Section 3.3 being invoked here?

AR-RC40: Please see AR-RC7 for details regarding the corrections to internal section referencing. Section 3.3 is intentionally being invoked. See page 7, lines 8 – 11.

RC41: In the conclusions on Pg. 16, it is stated that terrestrial warm INPs are abundant and that marine warm INPs are not evident, but there are warm INP data points reported for BBY in Figure 4. If these are not marine warm INPs, where do they come

from?

AR-RC41: The authors provide the following explanation for why warm INPs are ephemerally reported for BBY: "Both sites are downwind of marine particle sources for the entire storm and the cloud layers above each site receive significant airmass contribution from the marine boundary layer during all storm periods (Table 4). However, only the inland site shows warm INPs in precipitation during all periods (Fig 4a and Fig 6a,c). The only difference in airmass influence between the cloud layers over the two sites is that inflowing air to mixed phase clouds over the inland site (CZC) passes through the terrestrial boundary layer before arriving (Table 4). When warm INPs are present in coastal site precipitation, their presence can be explained by transport patterns and cloud altitude favorable for LRT aerosols to become injected at cloud top. Conversely, we cannot construct a similar alternate hypothesis explaining ephemeral injection of marine warm INPs into coastal site clouds." See page 15, lines 25 – 31 and page 16, line 1.

RC42: If the small, non-zero change in instantaneous element residence in the terrestrial boundary layer is really the driver of why the warm INP concentrations vary at CZC, then why do the INP concentrations not vary with the varying numbers shown in Table 4 – 6.2

AR-RC42: The authors note that the airmass source and its ambient warm INP concentration are only a single factor controlling the warm INP concentration of precipitation. Other kinematic and cloud microphysical processes may independently influence the final concentration measured in precipitation samples, thus the reader should not expect that the terrestrial boundary layer residence probability will covary with warm INP concentration in precipitation. The authors have added additional discussion to clarify this point, see AR-RC25.

Referee Two Technical Corrections: RC 43: On page 6, line 4, IVT is not yet defined. AR-RC43: Corrected as requested. See page 7, line 15.

RC 44: On page 9, Line 15, Reference to Martin et al. seems out of place.

AR-RC44: The enveloping passage was redacted in response to another comment. See AR-RC24.

RC 45: On page 9, line 18, Figure 15b reference is out of order.

AR-RC45: The enveloping passage was redacted in response to another comment. See AR-RC24.

RC 46: On page 12, line 17, The reference to Section 2d (2.4) does not seem right. Should this be 2.5?

AR-RC46: Yes, the referee is correct. This has been corrected, see page 11, line 22.

Please also note the supplement to this comment:
https://www.atmos-chem-phys-discuss.net/acp-2018-702/acp-2018-702-AC1-supplement.pdf